# Nanosensitizer-mediated augmentation of sonodynamic therapy efficacy and antitumor immunity

Yongjiang Li[1,8], Wei Chen[1,8], Yong Kang [2], Xueyan Zhen[1], Zhuoming Zhou[1], Chuang Liu [1], Shuying Chen[1], Xiangang Huang [1], Hai-Jun Liu[1], Seyoung Koo[1], Na Kong[1,3,4], Xiaoyuan Ji [2,4], Tian Xie[4,5,6,7] ✉ & Wei Tao [1] ✉

The dense stroma of desmoplastic tumor limits nanotherapeutic penetration and hampers the antitumor immune response. Here, we report a denaturation-and-penetration strategy and the use of tin monosulfide nanoparticles (SnSNPs) as nano-sonosensitizers that can overcome the stromal barrier for the management of desmoplastic triple-negative breast cancer (TNBC). SnSNPs possess a narrow bandgap (1.18 eV), allowing for efficient electron ($e^-$)-hole ($h^+$) pair separation to generate reactive oxygen species under US activation. More importantly, SnSNPs display mild photothermal properties that can in situ denature tumor collagen and facilitate deep penetration into the tumor mass upon near-infrared irradiation. This approach significantly enhances sonodynamic therapy (SDT) by SnSNPs and boosts antitumor immunity. In mouse models of malignant TNBC and hepatocellular carcinoma (HCC), the combination of robust SDT and enhanced cytotoxic T lymphocyte infiltration achieves remarkable anti-tumor efficacy. This study presents an innovative approach to enhance SDT and antitumor immunity using the denaturation-and-penetration strategy, offering a potential combined sono-immunotherapy approach for the cancer nanomedicine field.

Solid tumors, particularly desmoplastic tumors like triple-negative breast cancer (TNBC)[1], are characterized by dense stroma composed of abundant collagen fibers, hyaluronic acid, and fibroblasts. The stroma creates a barrier that restricts the delivery of therapeutics to the tumor parenchyma and hinders the deep penetration of nanomedicine into the tumor[2]. Moreover, the stromal barrier impedes the intra-tumoral infiltration of cytotoxic T lymphocytes (CTLs) capable of recognizing malignant cells and exerting killing effects[3]. Indeed, as highlighted in the FinHER trial, a higher level of tumor-infiltrating lymphocytes results in better survival outcomes and reduced risk of recurrence in primary TNBC[4]. Moreover, a meta-analysis study demonstrated improved survival in TNBC with high-level tumor infiltrating CD4[+] and CD8[+] T lymphocytes[5]. Despite the fact that abundant (more than 50%) tumor-infiltrated lymphocytes and stromal lymphocytes without direct contact with TNBC cells[6], these studies have highlighted the significant prognostic value of infiltrated lymphocytes in TNBC and suggested that overcoming the tumor-stromal barrier could improve prognosis. Previous nanomedicine-based strategies

[1]Center for Nanomedicine and Department of Anesthesiology, Brigham and Women's Hospital, Harvard Medical School, Boston, MA 02115, USA. [2]Academy of Medical Engineering and Translational Medicine, Medical College, Tianjin University, 300072 Tianjin, China. [3]Liangzhu Laboratory, Zhejiang University Medical Center, Hangzhou, China. [4]School of Pharmacy, Hangzhou Normal University, 311121 Hangzhou, Zhejiang, China. [5]Key Laboratory of Element Class Anti-Cancer Chinese Medicines, Hangzhou Normal University, 311121 Hangzhou, Zhejiang, China. [6]Engineering Laboratory of Development and Application of Traditional Chinese Medicines, Hangzhou Normal University, 311121 Hangzhou, Zhejiang, China. [7]Collaborative Innovation Center of Traditional Chinese Medicines of Zhejiang Province, Hangzhou Normal University, 311121 Hangzhou, Zhejiang, China. [8]These authors contributed equally: Yongjiang Li, Wei Chen. ✉e-mail: tianxie@hznu.edu.cn; wtao@bwh.harvard.edu

focused on disrupting the tumor stroma by directly targeting cancer-associated fibroblasts (CAFs) or remodeling the tumor extracellular matrix (ECM)[7]. For example, targeting fibroblast activation protein with pFap DNA vaccines can improve chemotherapy outcomes by increasing intratumoral drug uptake[8]; degradation of tumor ECM with hyaluronidase[9] or collagenase[10] can improve intratumoral diffusion of nanoparticles (NPs) and drugs. However, these strategies pose a risk of fibroblast and ECM disruption to normal tissues due to their limited tumor-targeting efficiency. Therefore, developing a strategy that can facilitate in situ intratumoral penetration of therapeutics and lymphocytes is of great significance for desmoplastic tumor management.

Sonosensitizer-assisted sonodynamic therapy (SDT) is an emerging non-invasive and in situ activable approach for tumor treatment[11,12]. Ultrasound (US) offers several advantages for therapeutic applications, including high controllability, non-invasiveness, and, more importantly, deep tissue penetration capability (on the order of centimeters) for in situ treatment[13–15]. Remarkably, several ongoing clinical trials (NCT05362409; NCT05580328) are currently assessing the therapeutic efficacy of sonosensitizer-based SDT, either independently or in combination with photodynamic therapy (PDT), for the treatment of tumors. Indeed, these trials have highlighted the significant clinical promise of tumor SDT. Like photosensitizers used in PDT, sonosensitizer can induce the separation of electron ($e^-$)–hole ($h^+$) pairs under US irradiation. The released energy and $e^-$ and $h^+$ can react with surrounding $O_2$ or $H_2O$ to produce cytotoxic reactive oxygen species (ROS), leading to cell death[16,17]. Notably, the efficiency of ROS generation, which determines the SDT effects, is associated with the bandgap size between the valence and conduction bands of sonosensitizers. For example, titanium dioxide ($TiO_2$), a representative sonosensitizer, has a relatively low ROS generation owing to its wide bandgap (3.2 eV) and fast combination of $e^-$ and $h^+$[18]. Despite extensive efforts to develop NP-based sonosensitizers, such as sodium molybdenum bronze NPs with a bandgap of 2.7 eV[19] and $BiVO_4$ with a bandgap of 2.5 eV[20], the challenge remains in efficiently generating ($e^-$)–hole ($h^+$) pairs in these nanomaterials due to their wide bandgap. Therefore, the development of innovative sonosensitizers with a narrow bandgap is highly desirable for the efficient generation of cytotoxic ROS for enhancing SDT[11,14]. In addition to the in situ activable features of sonosensitizers for SDT, using such sonosensitizers to overcome the dense stromal barrier and penetrate deeply into tumors is particularly advantageous for improving tumor treatment efficacy.

Tin monosulfide (SnS) is a binary compound in the IV-VI group that exhibits potential for various biomedical applications[21]. SnS, like black phosphorous, has a two-layer orthorhombic crystal structure with low symmetry and high stability. Notably, SnS has a narrow bandgap (ranging from 1.07 eV[22] to 1.81 eV[23]), which enables it to be activated by external stimuli for photo-mediated[24] and potentially sono-mediated biomedical applications. In this study, we successfully exfoliated SnS powder into SnS nanoparticles (SnSNPs) using a robust top-down liquid-phase exfoliation technique[25–29]. SnSNPs showed high ROS generation capability under US irradiation due to their narrow bandgap ($E_g$ = 1.18 eV). Moreover, SnSNPs possess near-infrared (NIR) absorption and mild photothermal effects, which can enhance tumor oxygen supply[30] and denature the tumor collagen[31], facilitating intratumoral penetration of SnSNPs and improving the SDT effects. Given the stromal barrier challenge and the paradigm and properties of SnSNPs, we designed a denaturation-and-penetration strategy to enhance SDT and antitumor immunity for the treatment of desmoplastic TNBC. Due to their mild photothermal effect, SnSNPs can denature tumor collagen, and penetrate deeply into the tumor, leading to tumor cell death through ROS generation under US irradiation. Meanwhile, the denaturation of tumor collagen stroma can enhance the infiltration of CTLs, which recognize tumor antigens, generate antitumor immunity, and kill the residual tumor cells[32]. We observed significantly improved therapeutic efficacy of SDT and antitumor

immunity with the treatment strategy (Fig. 1). The SnSNPs-mediated therapy completely eradicated the tumor without recurrence in a mouse model of orthotopic TNBC and effectively suppressed the tumor growth in a mouse model of orthotopic hepatocellular carcinoma (HCC). Importantly, SnSNPs demonstrated high biocompatibility and did not induce any toxicity in the treated tumor-bearing mice. This work highlights the potential of the nanoparticle-mediated denaturation-and-penetration strategy to achieve robust tumor therapy based on enhanced SDT and antitumor immunity.

## Results

### Synthesis and characterization of SnSNPs and SnSNPs@PEG
SnSNPs were synthesized via a robust liquid-phase exfoliation strategy. As shown in transmission electron microscope (TEM) image (Fig. 2a), the average size of SnSNPs was approximately 14.8 ± 2.5 nm. The X-ray diffraction (XRD) showed the orthorhombic crystal structure of SnSNPs (ICDD # 039-0354) (Fig. 2b). Additionally, the high-resolution TEM image showed the clear crystal structure that the spacing of the lattice is 0.28 nm, which is consistent with the spacing of (040) planes of SnS (Fig. 2c). The X-ray photoelectron spectroscopy (XPS) survey spectrum showed the chemical composition of SnSNPs (Fig. S2). To improve the colloidal stability, we coated the SnSNPs with 1,2-distearoyl-sn-glycero-3-phosphoethanolamine-N-[methoxy(polyethylene glycol)] (DSPE-PEG), resulting in a formation of SnSNPs@PEG. The coating with PEG of SnSNPs was confirmed through thermogravimetric analysis (TGA) and Fourier transform infrared (FTIR). The weight loss difference between SnSNPs and SnSNPs@PEG after heating to 800 °C was found to be 28 wt% (Fig. 2d), suggesting efficient PEG coating on the SnSNPs surface. Additionally, FTIR analysis of SnSNPs@PEG revealed the appearance of absorption bands at 2893, 1748, 1473 and 1346, and 1115 cm$^{-1}$, corresponding to C–H stretching, C = O stretching, C–H bending, and C = O stretching, respectively, demonstrating the successful PEG modification on the SnSNPs surface (Fig. 2e). Importantly, dynamic light scattering (DLS) analysis revealed that the hydrodynamic size of SnSNPs@PEG in phosphate-buffered saline (PBS) solution was 32.6 ± 1.8 nm, with a polydispersity index of 0.264 (Figs. 2f, S1a). Compared with bare SnSNPs, the successful PEG coating can significantly improve the colloidal stability of SnSNPs in biologically relevant environments such as RPMI-1640 medium and PBS solution, with no significant increase in size (Fig. S1). Additionally, the zeta potential of SnSNPs@PEG was −16.1 ± 1.0 mV and no noticeable change was observed after storage for five days (Fig. S1a). The improvement in stability makes SnSNPs@PEG more advantageous for biological applications.

### Sonodynamic performance of SnSNPs@PEG
To investigate the potential of SnSNPs for the application of SDT, we first measured their optical bandgap. Using the optical absorbance spectrum of solid SnSNPs and a Tauc plot of the Kubelka−Munk function (Fig. 3a), the optical bandgap ($E_g$) of SnSNPs was estimated to be 1.18 eV (Fig. 3a), consistent with the previous characterization of SnS as a semiconductor[22,23]. Next, to investigate the sonodynamic performance of SnSNPs@PEG, we used 1,3-diphenylisobenzofuran (DPBF) and methylene blue (MB) as probes to detect the generation of $^1O_2$ and ·OH, respectively (Fig. 3b, e). First, the solution containing SnSNPs@PEG and DPBF was exposed to the US (1 MHz, 1 Wcm$^{-2}$, 50% duty cycle). The absorbance peak of DPBF at 410 nm in the UV−Vis spectra decreased in the US exposure time-dependent manner (Fig. 3c), demonstrating that $^1O_2$ generated by US-triggered SnSNPs@PEG oxidized DPBF into colorless 1,2-dibenzoylbenzene over time (Fig. 3d). Comparatively, negligible DPBF degradation was observed in all control groups including US and SnSNPs@PEG only (Fig. 3d; Fig. S3).

Subsequently, the solution containing MB and SnSNPs@PEG was exposed to the US trigger (1 MHz, 2 W cm$^{-2}$, 50% duty cycle) to detect

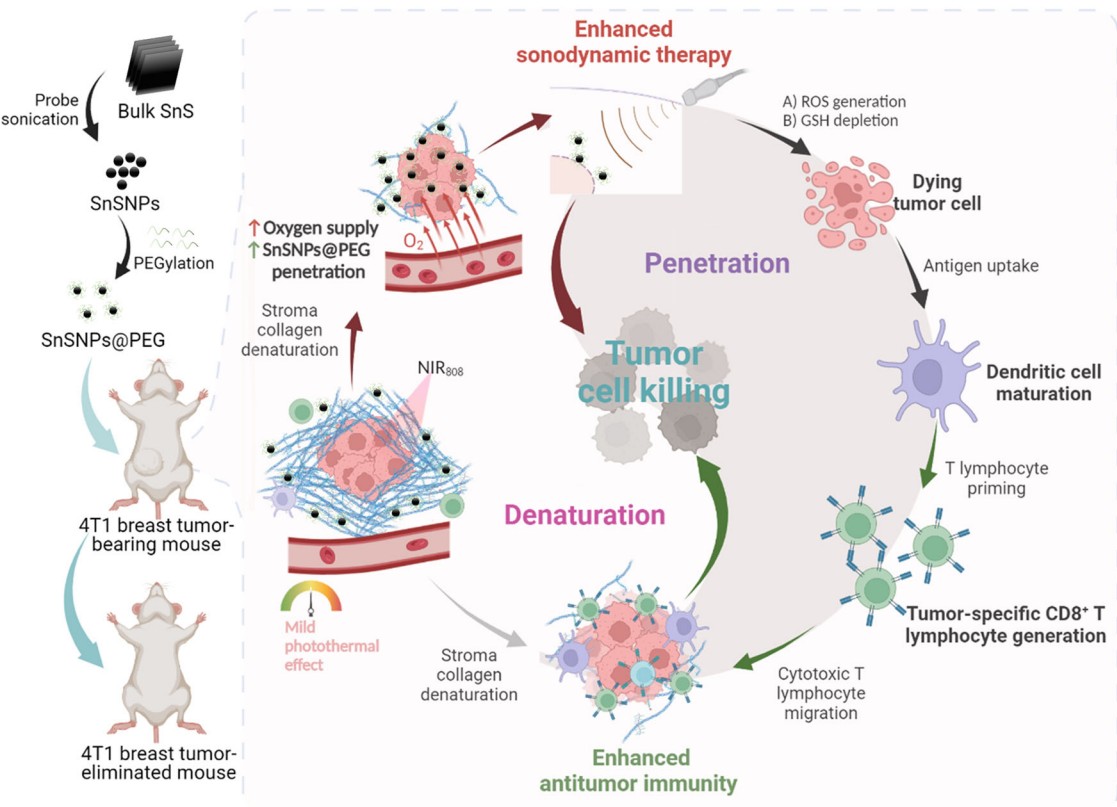

**Fig. 1 | Schematic illustration of the preparation of polyethylene glycol (PEG)-coated SnS nanoparticles (SnSNPs@PEG) and their denaturation-and-penetration strategy for enhanced SDT.** SnSNPs@PEG were prepared via the liquid-phase exfoliation method and subsequent PEGylation. The mild photothermal effect of SnSNPs@PEG increases tumor oxygen supply and denatures tumor collagen, promoting the penetration of SnSNPs@PEG to enhance SDT and kill tumor cells. This approach also boosts antitumor immunity by activating cytotoxic T lymphocytes that can infiltrate into the tumor overcoming the stromal barrier and further disrupting tumors for antitumor effects. The figure was created with Biorender.com.

the generation of ·OH (Fig. 3e). The absorbance peak of MB at 668 nm in the UV−Vis spectra gradually decreased with increasing US exposure time (Fig. 3f). Moreover, the peak intensity diminished more significantly with the addition of $H_2O_2$ (Fig. 3g). These results proved that ·OH generated by US-triggered SnSNPs@PEG reacted with MB and turned the solution colorless over time (Fig. 3h). In addition, it should be noted that the absorbance peak of MB was also decreased when the solution was exposed to US only (Fig. S4a). Furthermore, the decreasing trend observed with US exposure was amplified by adding $H_2O_2$ (Fig. S4b). Remarkably, the absorbance peak at 668 nm was prominently lower after US irradiation with the addition of SnSNPs@PEG (Fig. 3h). In contrast, no decrease in absorbance peaks was observed for other control groups without the US exposure, including SnSNPs@PEG, SnSNPs@PEG + $H_2O_2$, and SnSNPs@PEG + $H_2O_2$ + $NIR_{808}$ (Fig. S5). These results demonstrated desirable $^1O_2$ and ·OH generation from SnSNPs@PEG by US-triggering, indicating the high sonosensitizer potential of SnSNPs@PEG for SDT.

As a scavenger for ROS in cells, glutathione (GSH) can limit the effectiveness of US-mediated ROS generation. Consequently, the US-mediated depletion of GSH has the potential to improve the antitumor efficacy. To detect GSH depletion, we used 5,5-dithio-bis-(2-nitrobenzoic acid) (DTNB) as a probe. Upon reacting with h+ from US-activated SnSNPs@PEG, GSH is converted to its oxidized form, glutathione disulfide (GSSG). In this study, GSH was mixed with SnSNPs@PEG and exposed to US (1 MHz, 2 W cm$^{-2}$, 50% duty cycle). After irradiation, the remaining GSH reacted with DTNB to produce a yellow product (GSH-DTNB), which has an absorption peak at 412 nm (Fig. 3i). The absorption peak of GSH-DTNB at 412 nm decreased over time when SnSNPs were exposed to US irradiation (Fig. 3j). In contrast,

no depletion of GSH was observed for all other control groups (Figure S6), and US irradiation without SnSNPs@PEG showed no effects (Fig. 3k). These results demonstrated that US-irradiated SnSNPs@PEG generated ROS that can deplete GSH in a time-dependent manner. Collectively, we demonstrated that SnSNPs@PEG, with a narrow bandgap, can efficiently separate e− and h+ under US activation to generate $^1O_2$ and ·OH under US activation (Fig. 3l).

**Photothermal performance of SnSNPs@PEG**
The dark color of SnSNPs@PEG suggested it has strong absorbance in the visible and near-infrared (NIR) region and inspired us to investigate its photothermal performance for collagen denaturation (Fig. 4a). The UV−Vis−NIR absorption spectra of SnSNPs@PEG revealed a broad, flat, and strong concentration-dependent absorption from 400 nm to 900 nm (Fig. S7a), suggesting that SnSNPs@PEG could be a promising candidate for photothermal applications. Further, we evaluated the NIR-mediated photothermal performance of SnSNPs@PEG. When exposed to a $NIR_{808}$ laser, the temperature of the RPMI-1640 cell medium solution containing SnSNPs@PEG increased significantly over time, while the blank solution showed only a minimal temperature increase (Fig. 4a). Additionally, we found that the photothermal effect of SnSNPs@PEG was strongly dependent on its concentration (Fig. 4b) and the power density of the $NIR_{808}$ laser (Fig. 4a). To quantify the photothermal performance of SnSNPs@PEG, we calculated its photothermal conversion efficiency (PTCE), which was found to be 25.2% (Figs. S7b, S7c). Our findings suggest that SnSNPs@PEG exhibit a moderate PTCE, and a mild temperature increase is sufficient to greatly improve the SDT[30]. Next, to investigate the photothermal stability of

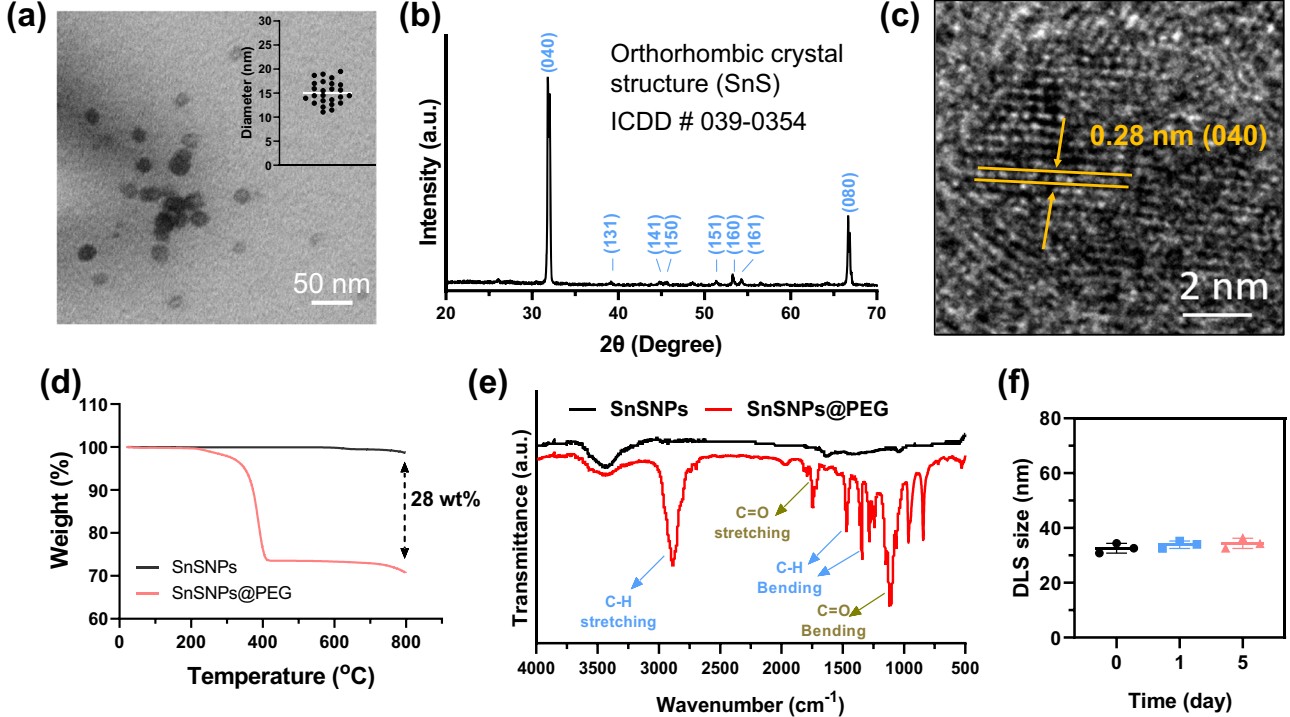

**Fig. 2 | Physicochemical characterization of SnSNPs and SnSNPs@PEG.**
**a** Transmission electron microscopy (TEM) image of SnSNPs and size measurement. Experiment was repeated two times independently with similar results.
**b** XRD pattern of SnSNPs. **c** High-resolution TEM image of SnSNPs. Experiment was repeated two times independently with similar results. **d** TGA of SnSNPs and SnSNPs@PEG. **e** FTIR spectra of SnSNPs and SnSNPs@PEG. **f** Hydrodynamic size of SnSNPs@PEG on different days measured by dynamic light scattering, data are presented as mean ± SD ($n = 3$).

SnSNPs@PEG, RPMI-1640 cell medium containing SnSNPs@PEG was repeatedly exposed to the $NIR_{808}$ laser for 10 min (ON) followed by naturally cooling down to room temperature (OFF). The results showed that SnSNPs@PEG exhibited stable photothermal effects without significant change in peak temperature for five ON/OFF cycles (Fig. S7d), indicating that SnSNPs@PEG has high photothermal stability under biological environments. Taken together, these results demonstrate that SnSNPs@PEG has an effective, mild, and stable heating ability under $NIR_{808}$ irradiation, which has the potential to denature tumor stroma and enhance SDT.

## Biocompatibility and cellular uptake of SnSNPs@PEG

After confirming the sonodynamic and NIR-mediated photothermal performance of SnSNPs@PEG, we investigated its in vitro biocompatibility and cellular uptake. Firstly, we evaluated the biocompatibility of SnSNPs@PEG and observed no cytotoxicity even at a high concentration ($400\,\mu g\,mL^{-1}$) after 24 h and 48 h of incubation (Fig. S8). Subsequently, we examined the cellular uptake of SnSNPs@PEG using both fluorescence Cy5-labeled SnSNPs@PEG (SnSNPs@PEG-Cy5) under a confocal laser scanning microscope and direct observation of the dark SnSNPs@PEG under an optical microscope. A significant and time-dependent cellular uptake was observed for both SnSNPs@PEG and SnSNPs@PEG-Cy5 by 4T1 cells (Fig. 4c). SnSNPs@PEG was observed in cells after 2 h of incubation, and a time-dependent uptake was demonstrated, which suggested that 4T1 cells can efficiently uptake SnSNPs@PEG (Fig. 4c). Importantly, we observed no change in cell morphology after substantial uptake of SnSNPs@PEG (Fig. S9), indicating high biocompatibility of SnSNPs@PEG without an external trigger.

## In vitro SDT of SnSNPs@PEG

With SnSNPs@PEG exhibiting high biocompatibility and efficient cellular uptake, we proceeded to investigate its SDT performance on 4T1 cells in vitro (Fig. 4d). It should be noted that for in vitro SDT, we used a low-intensity US to avoid US-induced cell detachment from the tissue culture plate as this effect can significantly influence the results of cell viability assay. The cells were incubated with SnSNPs@PEG for 24 h to ensure sufficient uptake, followed by exposure to $NIR_{808}$ ($1.0\,W\,cm^{-2}$) and low-intensity US ($0.3\,W\,cm^{-2}$, 1 MHz, 50% duty cycle). The viability of 4T1 cells was reduced to 68% and 49% after treatment with SnSNPs@PEG and US (SnSNPs@PEG + US, G5), and SnSNPs@PEG combined with NIR and US (SnSNPs@PEG + $NIR_{808}$ + US, G6), respectively (Fig. 4e). The results were consistent with observations of cells under a microscope (Fig. S10) and cell colony staining by crystal violet (Fig. 4f, g).

The effectiveness of SnSNPs@PEG in SDT against 4T1 cells was further confirmed through the Calcein-AM/PI co-staining assay (Fig. 4h). Compared with the control groups (PBS, US only, NIR + US, and SnSNPs@PEG only), 4T1 cells treated with SnSNPs@PEG followed by US irradiation exhibited strong red fluorescence, indicating dying cells, and less green fluorescence of living cells. Notably, the group treated with $NIR_{808}$ irradiation and US exposure showed the most significant therapeutic effects (Fig. 4h). Collectively, these findings demonstrate the remarkable antitumor potential of the enhanced SDT by SnSNPs@PEG.

Next, we investigated the intracellular mechanism of US-mediated tumor cell-killing effects of SnSNPs@PEG by monitoring intracellular ROS generation using a 2,7-dichlorofluorescein diacetate ($H_2$DCF-DA) staining assay. We found that low-intensity US exposure combined with SnSNPs@PEG resulted in strong fluorescence and ROS generation (Fig. 4i). Remarkably, the group treated with SnSNPs@PEG followed by $NIR_{808}$ and US showed the most significant green fluorescence and ROS generation, while control groups (PBS, US, NIR + US, SnSNPs@PEG only) showed minimal fluorescence intensity (Fig. 4i, j). These results demonstrated that SnSNPs@PEG triggered by the US could efficiently generate ROS in 4T1 cells.

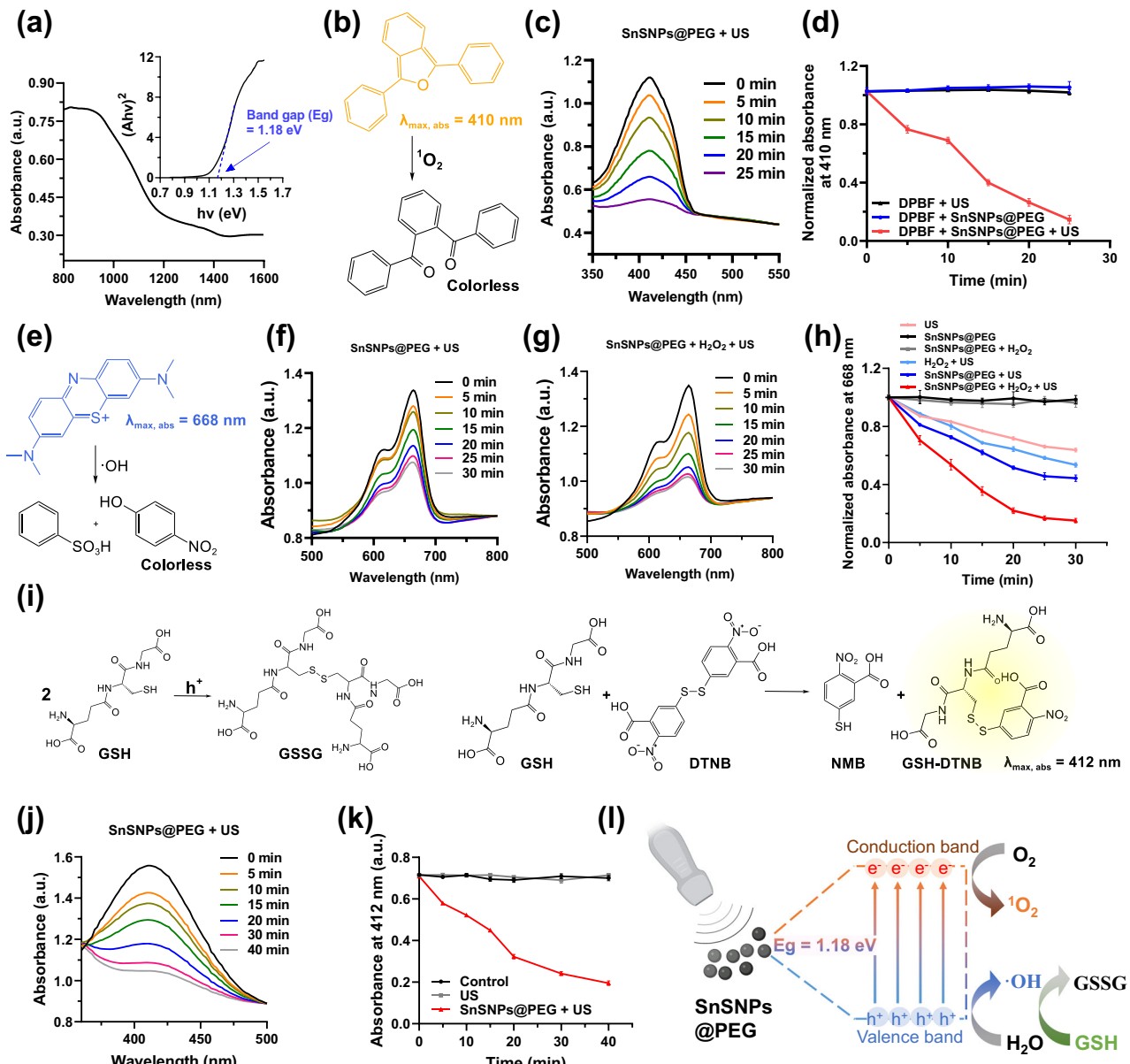

**Fig. 3 | Sonodynamic performance of SnSNPs@PEG. a** UV–Vis-NIR diffuse reflectance spectra of SnSNPs. Inset graph shows the corresponding optical bandgap ($E_g$) of SnSNPs calculated by the Kubelka–Munk equation. **b, c** Time-dependent oxidation of DPBF by $^1O_2$ generated from US (1 MHz, 1 Wcm$^{-2}$, 50% duty cycle)-triggered SnSNPs@PEG. **d** Comparison of DPBF oxidation by US only, SnSNPs@PEG, and SnSNPs@PEG + US. Data are from three independent experiments and are presented as mean ± SD ($n = 3$). **e–g** Time-dependent degradation of MB by ·OH generated from US (1 MHz, 2 W cm$^{-2}$, 50% duty cycle)-triggered SnSNPs@PEG. **h** Comparison of degradation of MB by SnSNPs@PEG under different treatments. Data from three independent experiments and are presented as mean ± SD ($n = 3$). **i, j** Time-dependent degradation of GSH by h$^+$ generated from US (1 MHz, 2 W cm$^{-2}$, 50% duty cycle)-triggered SnSNPs@PEG. **k** Comparison of degradation of GSH under different treatments, data are presented as mean ± SD ($n = 4$). **l** Mechanism of sonodynamic performance of SnSNPs@PEG under US trigger. GSH, glutathione; GSSG, glutathione disulfide. Illustration was created with BioRender.com.

## In vivo biodistribution and mild photothermal effects of SnSNPs@PEG

The biodistribution and mild photothermal effects of SnSNPs@PEG were evaluated in orthotopic 4T1 tumor-bearing mice. Following intravenous (i.v.) injection, SnSNPs@PEG-Cy5 exhibited significant accumulation at the tumor site (Fig. 5a, b), as demonstrated by strong fluorescence intensity, while the free DSPE-PEG-Cy5 control showed minimal fluorescence (Fig. 5a, b). These results confirmed the accumulation ability of SnSNPs@PEG to the tumor via the enhanced permeability and retention (EPR) effects[33]. Furthermore, the in vivo photothermal effects of SnSNPs@PEG were investigated by monitoring the temperature at the tumor site under NIR$_{808}$ irradiation 12 h

after i.v. injection of SnSNPs@PEG. The temperature at the tumor site increased quickly to 44 °C within 4 min of NIR$_{808}$ irradiation (1.0 W cm$^{-2}$) (Fig. 5c, d, S11), while NIR$_{808}$ irradiation without SnSNPs@PEG only increased the temperature by 3.5 °C (Fig. 5c, d). These results indicate the mild yet effective photothermal capability of SnSNPs@PEG.

### In vivo enhanced SDT of SnSNPs@PEG

Encouraged by the enhanced SDT of SnSNPs@PEG in in vitro studies, we next investigated the in vivo antitumor efficacy of SnSNPs@PEG on a mouse model of orthotopic TNBC (Fig. 6a). To evaluate the in vivo therapeutic efficacy of SnSNPs@PEG, we

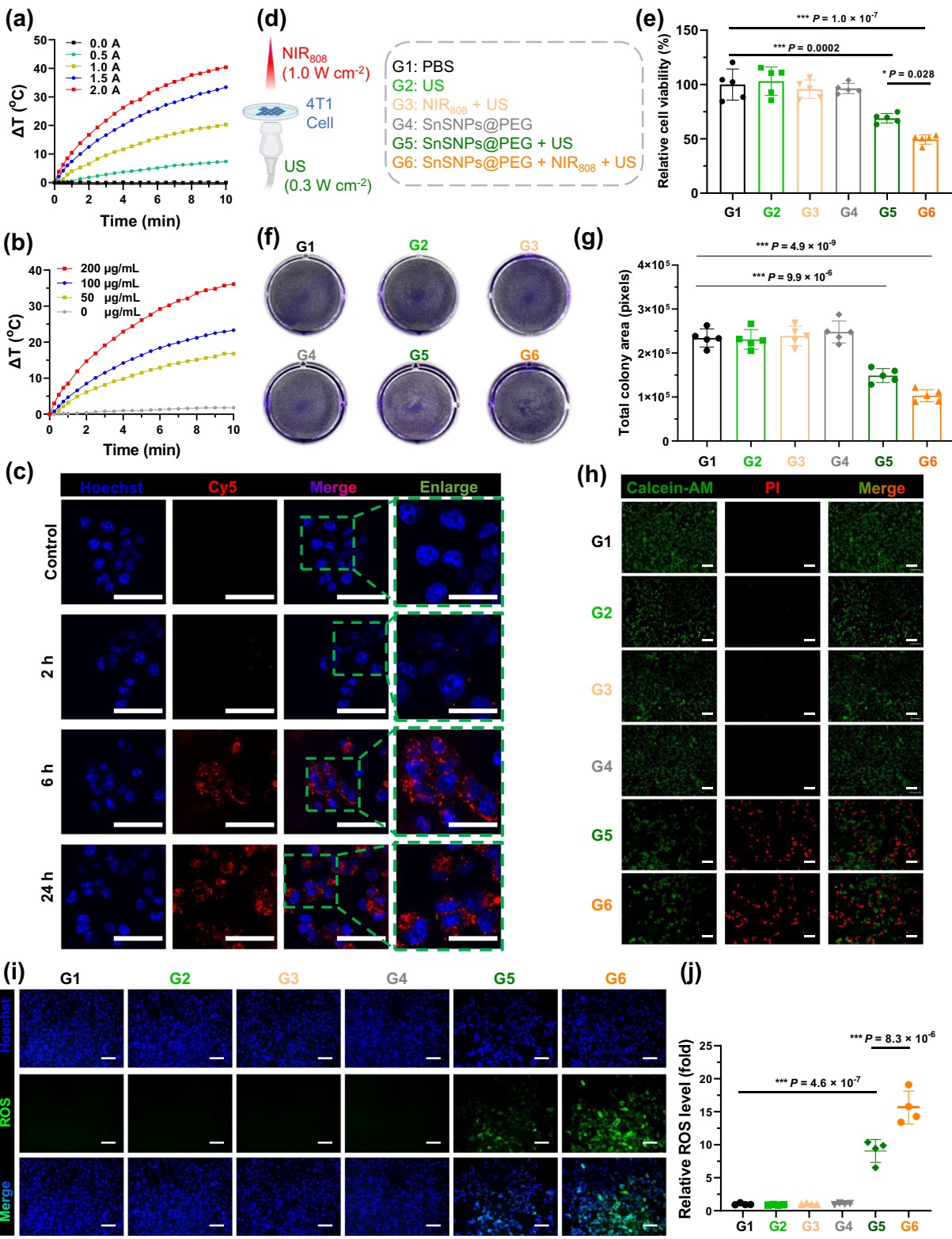

established the luciferase-expressing 4T1 cell (4T1-luc) tumor-bearing mice in which the tumor expansion can be noninvasively monitored via bioluminescence imaging. The 4T1-luc tumor-bearing mice were randomly divided into six groups: G1) PBS; G2) $NIR_{808}$ + US; G3) SnSNPs@PEG; G4) SnSNPs@PEG + US; G5) SnSNPs@PEG + $NIR_{808}$ + US; G6) SnSNPs@PEG + $NIR_{808}$ + US × 2. For the therapeutic assay, 24 h after i.v. injection of PBS or SnSNPs@PEG, the tumor regions of G2, G5, and G6 were exposed to

$NIR_{808}$ for 10 min (1 W cm$^{-2}$), and G2, G4, G5, and G6 were exposed to US irradiation for 10 min (5 min per cycle, two cycles, 1 MHz, 2 W cm$^{-2}$, 50% duty cycle) (Fig. 6a). First, as shown in the bioluminescence imaging (Fig. 6b, c), the $NIR_{808}$ + US (G2) and SnSNPs@PEG (G3) groups exhibited a negligible effect on tumor burden; the tumor luminescence intensity at day 16 of G1, G2, and G3 were 37.8 ± 13.5, 40.2 ± 13.6, and 39.1 ± 10.6 fold, respectively (Fig. S12a), verifying the biocompatibility of the combination of

**Fig. 4 | In vitro photothermal ability and SDT effect of SnSNPs@PEG. a** Power density-dependent photothermal performance of SnSNPs@PEG (200 μg mL⁻¹) irradiated by an 808 nm near-infrared (NIR) laser. **b** Concentration-dependent photothermal performance of SnSNPs@PEG irradiated by an 808 nm NIR laser (2.0 W cm⁻²). **c** Confocal laser scanning microscope images showing the cellular uptake of SnSNPs@PEG-Cy5 by 4T1 cells; scale bars were 50 and 25 μm respectively for merged images and enlarged views. Experiment was repeated two times independently with similar results. **d** Illustration of the in vitro SnSNPs@PEG-mediated SDT and groups of the assay. Illustration was created with BioRender.com. US, ultrasound. **e** Viability of 4T1 cells after SnSNPs@PEG-mediated SDT (200 μg mL⁻¹). Data are from five independent samples ($n = 5$). **f** Crystal violet staining showing the colony of cells after SnSNPs@PEG-mediated SDT (200 μg mL⁻¹). Data are from independent experiments ($n = 5$). **g** Analysis of the total colony area of cells after SnSNPs@PEG-mediated SDT. Data are from five independent samples ($n = 5$). **h** Calcein-acetoxymethyl ester (Calcein-AM)/Propidium iodide (PI) staining showing the live (green) and dead (red) 4T1 cells after SnSNPs@PEG-mediated SDT, scale bar = 100 μm. Experiment was repeated three times independently with similar results. **i** Fluorescence microscope images showing the intracellular reactive oxygen species (ROS) level detected by $H_2$DCF-DA probe (green), scale bar = 100 μm. **j** Analysis of the intensity of green fluorescence showing the relative ROS level ($n = 4$). Data are presented as mean ± SD. Multiple comparisons among groups were performed by one-way ANOVA. *$P < 0.05$, **$P < 0.01$, ***$P < 0.001$.

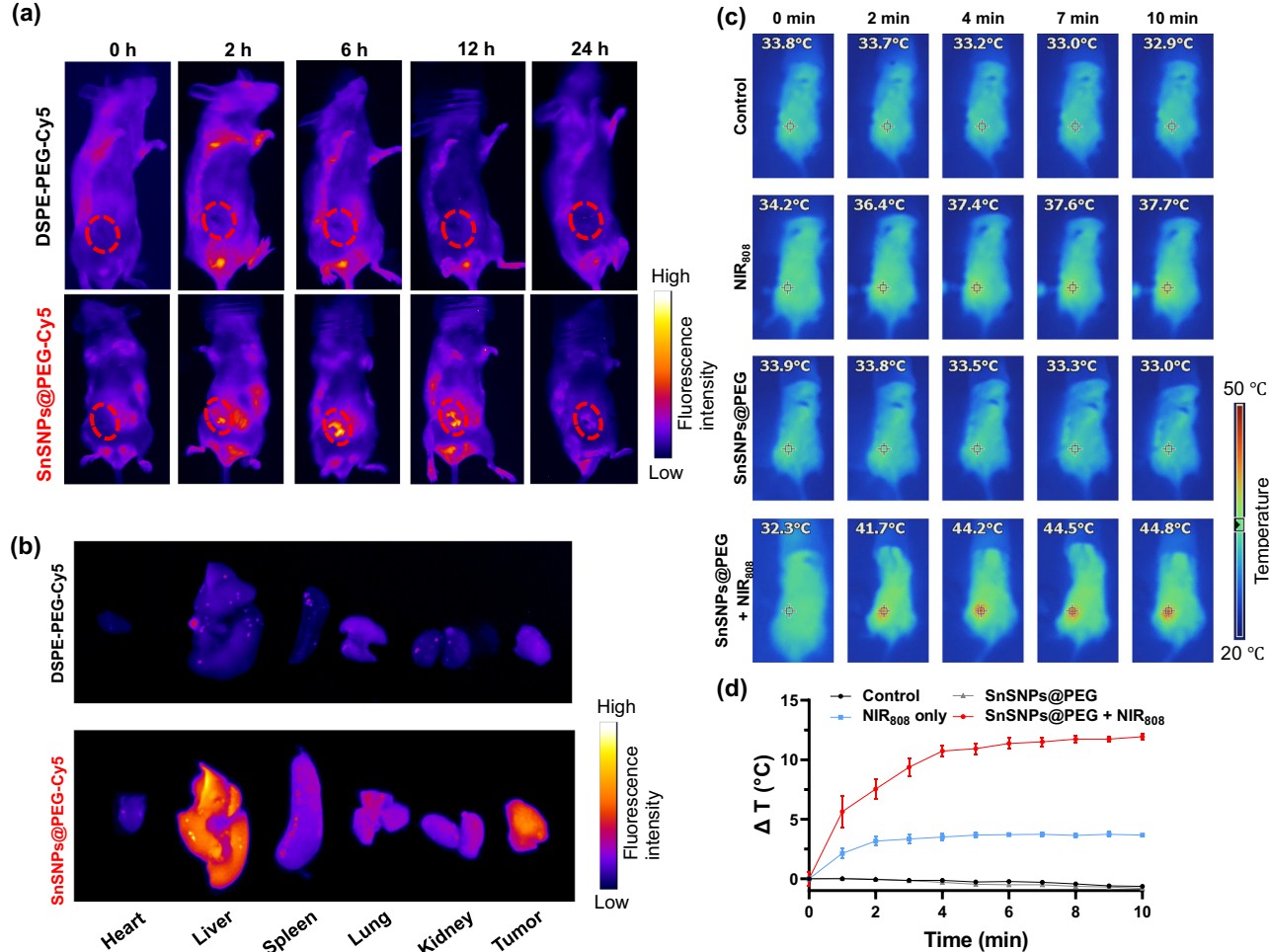

**Fig. 5 | Biodistribution and in vivo photothermal effect of SnSNPs@PEG. a** In vivo biodistribution of SnSNPs@PEG-Cy5 in 4T1 tumor-bearing mice. The red dotted circles indicate the tumor site. **b** Ex vivo biodistribution of SnSNPs@PEG-Cy5 in major organs and 4T1 tumors 24 h after administration. **c** IR thermal images of 4T1 tumor-bearing mice at different time points after various treatments. The tumor sites irradiated by NIR are marked by circles (power density: 1 W cm⁻²). **d** Time-dependent temperature increase profiles of tumor sites after various treatments were measured by IR thermal images ($n = 3$). Data are presented as mean ± SD.

these two external stimuli and SnSNPs@PEG. Moreover, the SnSNPs@PEG + US treatment (G4) showed moderate tumor growth suppression effects, with the average bioluminescence intensity at day 16 being 41.6% of that of the PBS control (12.7 ± 4.7-fold, Figs. 6b, c, S12). In addition, the combination with NIR₈₀₈ irradiation (G5) further enhanced the antitumor efficacy, as observed by bioluminescence (3.4 ± 0.7-fold, Figs. 6b, c, S12) and tumor volume measurements (Fig. 6f, g). Remarkably, two-time treatment of the combined therapy eradicated the tumor in mice without recurrence (Figs. 6b, c, f, g, S12b). Similarly, the tumor volume measurement showed that mice treated with one-time SnSNPs@PEG + NIR₈₀₈ + US increased by 2-fold, while the control increased by 8-fold (Fig. S13). This was supported by the image of excised tumors (Fig. 6d) and tumor weights (Fig. 6e), demonstrating similar results. Additionally, hematoxylin and eosin (H&E) and terminal deoxynucleotidyl transferase dUTP nick end labeling (TUNEL) staining confirmed notable cell death in tumors that received the combined treatment (Fig. 6g). Notably, there was no significant change in body weight during and after treatment (Fig. S14). Additionally, the size and weight of excised spleen

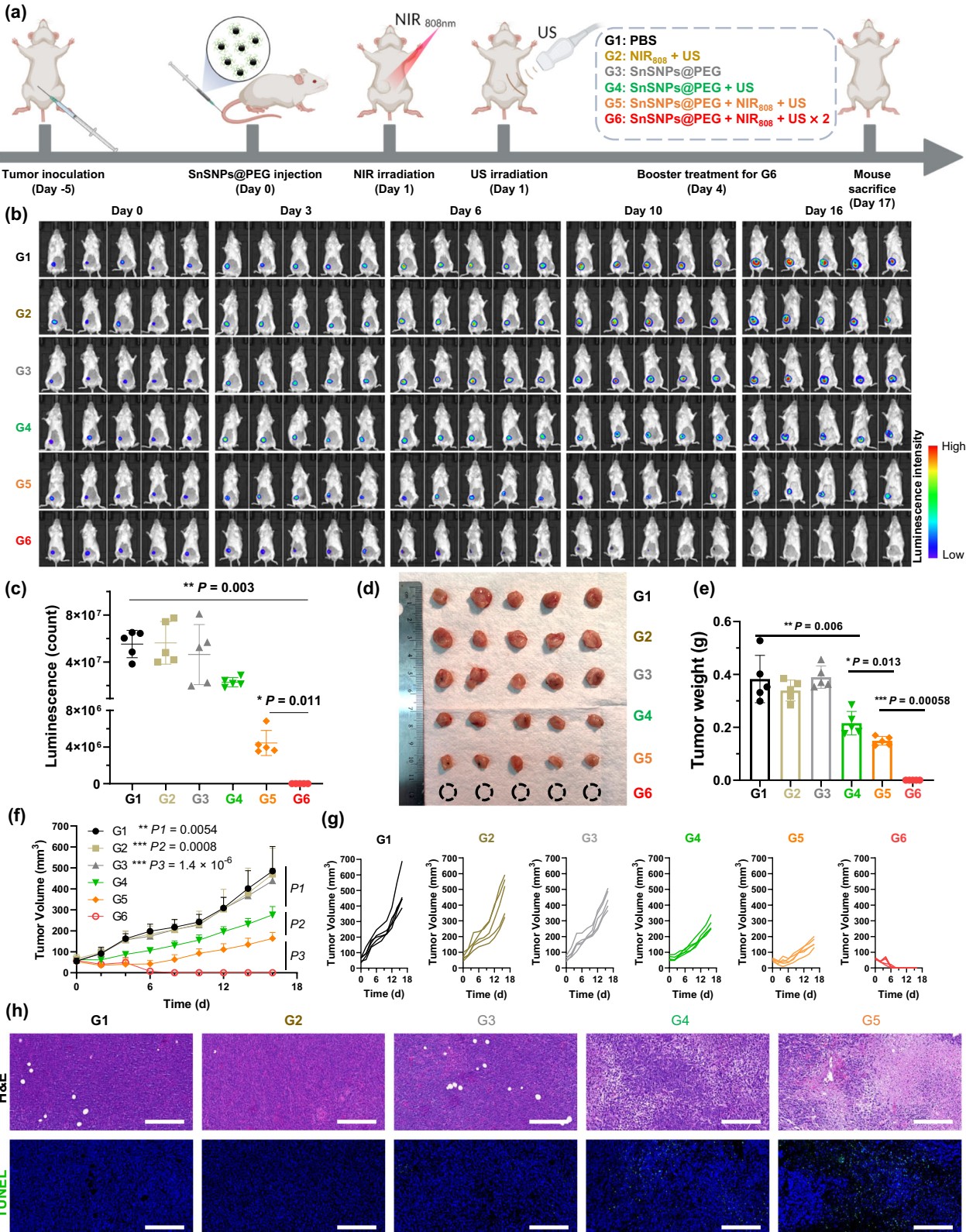

**Fig. 6 | Antitumor efficacy of SnSNPs@PEG-mediated SDT. a** Schedules of 4T1-tumor model establishment and SnSNPs@PEG-mediated treatment. Illustration was created with BioRender.com. **b** Bioluminescence images of orthotopic 4T1 tumor-bearing mice before, during and after various treatments. **c** Comparison of tumor luminescence intensity at Day 16 after various treatments (*n* = 5). **d** Photograph of excised tumors from orthotopic 4T1 tumor-bearing mice after various treatments, dash circle represents tumor were eradicated. **e** Weight of excised tumor from orthotopic 4T1 tumor-bearing mice after various treatments (*n* = 5). **f** Tumor growth curves of the orthotopic 4T1 tumor-bearing mice measured by a caliber (*n* = 5). **g** Individual tumor growth of tumor-bearing mouse in each group. **h** H&E staining and TUNEL fluorescence staining assays of excised 4T1 tumors after various treatments. Scale bar = 100 μm. Experiment was repeated three times independently with similar results. G6 is not included as tumors were eradicated. Data are presented as mean ± SD. Statistical analysis between two groups was performed by Student's *t* test. Multiple comparisons among groups were performed by one-way ANOVA. *P < 0.05, **P < 0.01, ***P < 0.001.

(Fig. S15) showed no significant toxicity during the treatment, demonstrating the high biocompatibility of the SnSNPs@PEG and the safety of the therapeutic strategy.

To assess the deep tissue penetration capability of SDT in enhanced tumor therapy, we developed an orthotopic HCC mouse model and evaluated the therapeutic effects (Fig. S16a). Our findings demonstrated significant anti-tumor efficacy of the SnSNPs@PEG-based therapeutic strategy. Treatment with SnSNPs@PEG + US slightly slowed tumor progression, while SnSNPs@PEG + NIR + US treatment showed enhanced therapeutic effects, as evidenced by bioluminescence imaging (Fig. S16b, S16c) and excised livers with tumor (Fig. S16d). Similarly, there was no significant change in mice body weight during the treatment (Fig. S16e). These results demonstrated that this SnSNPs@PEG-based therapeutic strategy is effective for deep-tissue tumor treatment.

### Denaturation of tumor collagen and enhanced intra-tumoral penetration of SnSNPs@PEG

We next studied whether the mild heating generated by NIR-irradiated SnSNPs@PEG can denature tumor collagen to enhance NP penetration. First, we used a channel filled with intact collagen I to test the effect of SnSNPs@PEG under $NIR_{808}$ irradiation on collagen denaturation and NP penetration (Fig. 7a). Both reservoirs were filled with PBS containing SnSNPs@PEG, with one side exposed to $NIR_{808}$ irradiation. The side exposed to $NIR_{808}$ irradiation showed elevated temperature and significant penetration of SnSNPs@PEG into the channel, as observed by microscope (Fig. 7a). Next, we evaluated whether this strategy could enhance tumor penetration in vivo. SnSNPs@PEG-Cy5 was intravenously injected into orthotopic 4T1 tumor-bearing mice, and 12 h later, the tumor site was exposed to $NIR_{808}$ irradiation, then harvested and sectioned for imaging. The tumor sections showed significantly enhanced penetration and distribution within the tumor for the SnSNPs@PEG + $NIR_{808}$ group compared to the no NIR group (Fig. 7b). The results from the analysis using inductively coupled plasma-mass spectrometry (ICP-MS) also showed a significantly higher concentration of Sn at the tumor site in the SnSNPs@PEG + $NIR_{808}$ group compared to the group without NIR treatment (Fig. 7c). In addition, Masson's trichrome staining showed decreased collagen fiber for the tumor treated with SnSNPs@PEG and $NIR_{808}$ irradiation (Fig. 7d). These results demonstrated that the SnSNPs@PEG-mediated mild photothermal effects could effectively denature the tumor collagen and enhance its intra-tumoral accumulation. Moreover, with improved tumor penetration of SnSNPs@PEG by $NIR_{808}$ irradiation, further US irradiation at the tumor site induced a more prominent in situ generation of ROS (Fig. 7e), nearly doubling the ROS level (Fig. S17) compared to the treatment of SnSNPs@PEG with US only. These findings suggest that SnSNPs@PEG combined with $NIR_{808}$ irradiation and US could serve as an effective strategy to improve intratumoral NP penetration and generate ROS for enhanced cancer therapy.

### Enhanced SDT by SnSNPs@PEG boosts antitumor immunity

Based on our denaturation-and-penetration strategy, enhanced accumulation of SnSNPs@PEG can improve the generation of ROS that can kill tumor cells and facilitate in situ antigen presentation for boosting antitumor immunity. Furthermore, the denaturation of tumor collagen facilitated the intra-tumoral infiltration of CTL, that have shown the ability for the clearance of residual tumor cells and the enhancement of antitumor immunity[34]. To confirm the improved penetration of immune cells in tumors, we performed flow cytometry analysis of total immune cells (CD45$^+$ cells), helper T lymphocytes (CD45$^+$CD3$^+$CD4$^+$ cells) and CTL (CD45$^+$CD3$^+$CD8$^+$ cells) in 4T1-tumors (Fig. S18) and in RIL-175 HCC (Fig. S19). Our results showed that SnSNPs@PEG treatment with $NIR_{808}$ and US irradiation significantly increased the percentage of total infiltrated immune cells (CD45$^+$) by 2.5-fold (from 3.6% to 9.1%), T lymphocytes (CD45$^+$CD3$^+$) by 2.6-fold (from 2.4% to 6.3%),

CD45$^+$CD3$^+$CD4$^+$ T lymphocytes by 3-fold (from 0.4% to 1.2%) and CD45$^+$CD3$^+$CD8$^+$ T lymphocytes by 5-fold (from 0.4% to 2.0%) in 4T1-tumors (Figs. 8a, S20). We also conducted a flow cytometry analysis of immune cells in HCC (Fig. S21). Similarly, in the orthotopic HCC mouse model, the treatment with SnSNPs@PEG with $NIR_{808}$ and US irradiation elevated the level of total CD45$^+$ cells (Fig. S22a), CD45$^+$CD3$^+$ cells (Fig. S22b), CD45$^+$CD3$^+$CD4$^+$ lymphocytes (Fig. S22c) and CD45$^+$CD3$^+$CD8$^+$ CTL (Fig. S22d) by 2.0-fold, 5.4-fold, 3.2-fold and 7.1-fold, respectively.

To evaluate the effect of our SnSNPs@PEG-based denaturation-and-penetration strategy on systemic antitumor immunity, we performed a flow cytometry analysis of immune cell populations in spleens and lymph nodes (Fig. S23). The percentage of CD45$^+$CD3$^+$CD8$^+$ CTL in total T lymphocytes was significantly increased by 11.7% in spleens (Figs. 8b, S23a) and 18.2% in lymph nodes (Figs. 8c, S23b) in 4T1-tumor-bearing mice treated with SnSNPs@PEG with US irradiation than controls. Moreover, the increase was elevated to 32.5% and 43.4% in spleens (Figs. 8b, S23a) and lymph nodes (Figs. 8c, S23b), respectively, in tumor-bearing mice treated with SnSNPs@PEG with combined $NIR_{808}$ and US irradiation. Collectively, these results demonstrated that the denaturation-and-penetration strategy boosted antitumor immunity by promoting the immune cell population, especially CTL infiltration into the tumor.

## Discussion

The diffusion of nanomedicine and the infiltration of lymphocytes into tumors has been challenging due to the tumor stromal barrier. To overcome the barrier and optimize the therapeutic effect, we take the advantage of noninvasive in situ activable sonosensitizer and design a nanoparticle-mediate denaturation-and-penetration strategy to enhance SDT and antitumor immunity. In this work, we reported the application of SnSNPs as sonosensitizers for the generation of a high yield of ROS. Owing to the narrow bandgap (1.18 eV), SnSNPs exhibited strong in situ ROS generation when exposed to US exposure. More importantly, the denaturation-and-penetration strategy remarkably enhanced the intra-tumoral accumulation of SnSNPs by overcoming the tumor stromal barrier based on the mild photothermal property of SnSNPs. Also, the denaturation of collagen in tumor stroma improved the infiltration of CTLs into the tumor, thereby recognizing and killing residual tumor cells after SDT. We found that the therapeutic strategy utilizing SnSNPs showed robust SDT effects and effectively boosted antitumor immunity, leading to the eradication of TNBC in mice without recurrence.

Despite the rapid progress of tumor immunotherapy, the therapeutic potential of immune-checkpoint inhibitors and chimeric antigen receptor (CAR) T cells remains limited to certain tumor types due to the immunosuppressive microenvironment established by the tumor stroma, which obstructs T cell infiltration. In TNBC, the tumor stroma is characterized by an excessive ECM produced by tumor-associated fibroblasts. The complex 3D meshwork of the ECM physically restricts the accumulation of NPs within the tumor. Recent studies have shown that the degradation of the ECM can not only facilitate the infiltration of nanomedicine[35] and immune cells[36], but also improve the efficacy of immune-checkpoint blockers (ICBs) such as anti-Programmed Death-1 antibody (αPD-1)[37]. In our study, the SnSNPs@PEG-mediated therapeutic strategy successfully facilitated the infiltration of CTLs into the tumor, leading to the elimination of residual tumor cells after SDT (Fig. 8). Our strategy not only demonstrated the effectiveness of combined sono-immunotherapy based on SnSNPs@PEG but also highlighted the therapeutic potential of combining the denaturation-and-penetration strategy with ICBs, such as αPD-1 and other immunotherapeutics for combinational tumor therapy. This strategy is particularly suitable for treating desmoplastic tumors, which are characterized by a dense ECM that acts as a defense against immune cells, like breast and lung cancers[38].

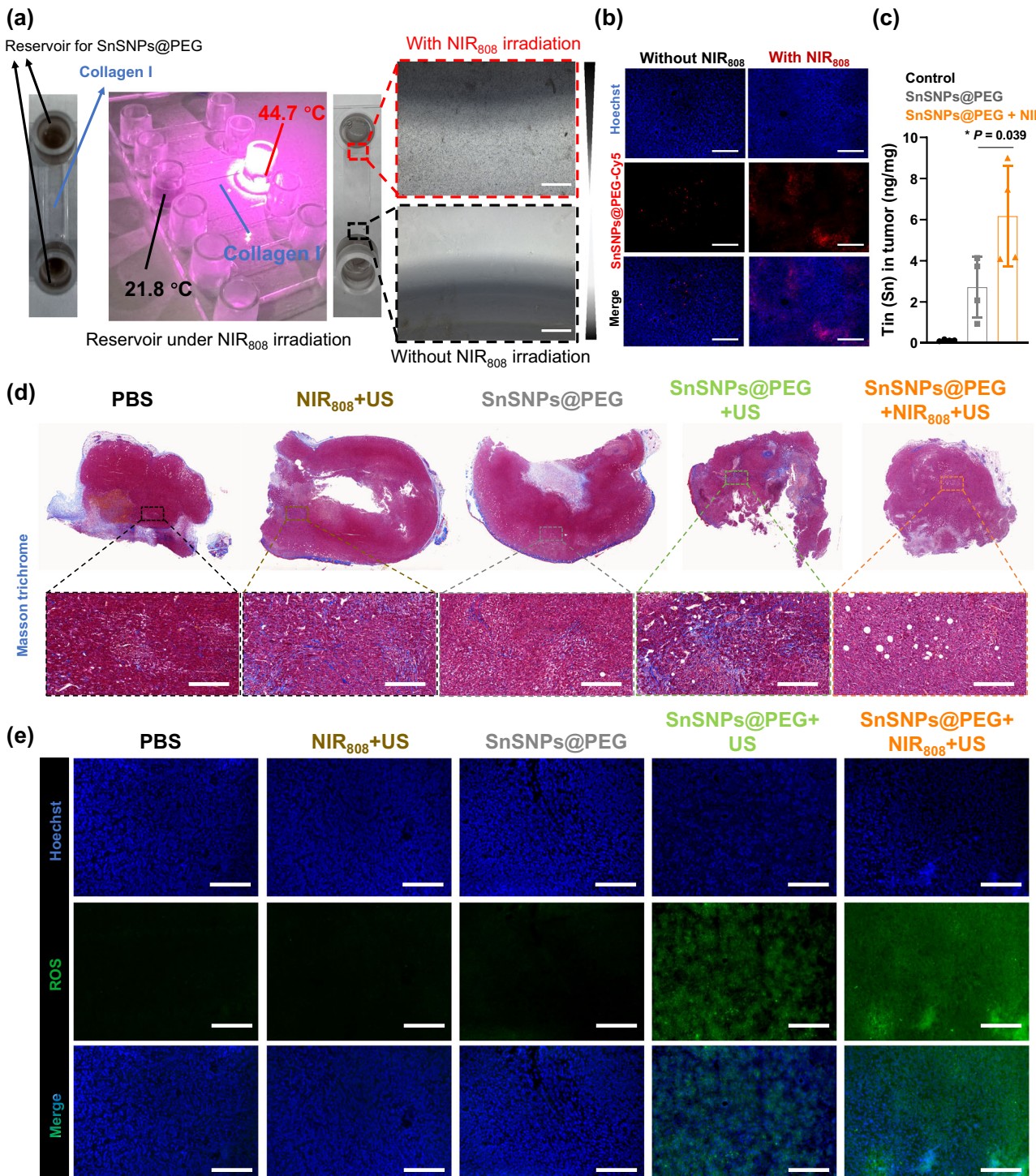

**Fig. 7 | NIR-mediated tumor collagen denaturation and SnSNPs@PEG penetration for enhanced SDT. a** Simulated assay of tumor collagen denaturation by μ-channel containing collagen I. Microscope images showing how $NIR_{808}$ irradiation facilitated the penetration of SnSNPs@PEG from the reservoir to the channel. Experiment was repeated three times independently with similar results. Scale bar = 100 μm. **b** Fluorescence microscope images of the tumor tissue sections showing the distribution of SnSNPs@PEG-Cy5 with or without $NIR_{808}$ irradiation. Scale bar = 100 μm. Experiment was repeated two times independently with similar results. **c** ICP-MS analysis of amount of Sn in 4T1-tumors with or without $NIR_{808}$ irradiation ($n = 4$). Data are presented as mean ± SD. Statistical analysis was performed by one-way ANOVA ($n = 4$). *$P < 0.05$. **d** Masson's trichrome staining of tumor tissue sections showing the content of collagen fibers after various treatments. Scale bar = 100 μm. Experiment was repeated two times independently with similar results. **e** Fluorescence microscope images of the tumor sections showing ROS level detected by $H_2$DCF-DA probe (Green) after various treatments. Scale bar = 100 μm. Experiment was repeated three times independently with similar results. PBS, phosphate-buffered saline; US, ultrasound; NIR, near-infrared.

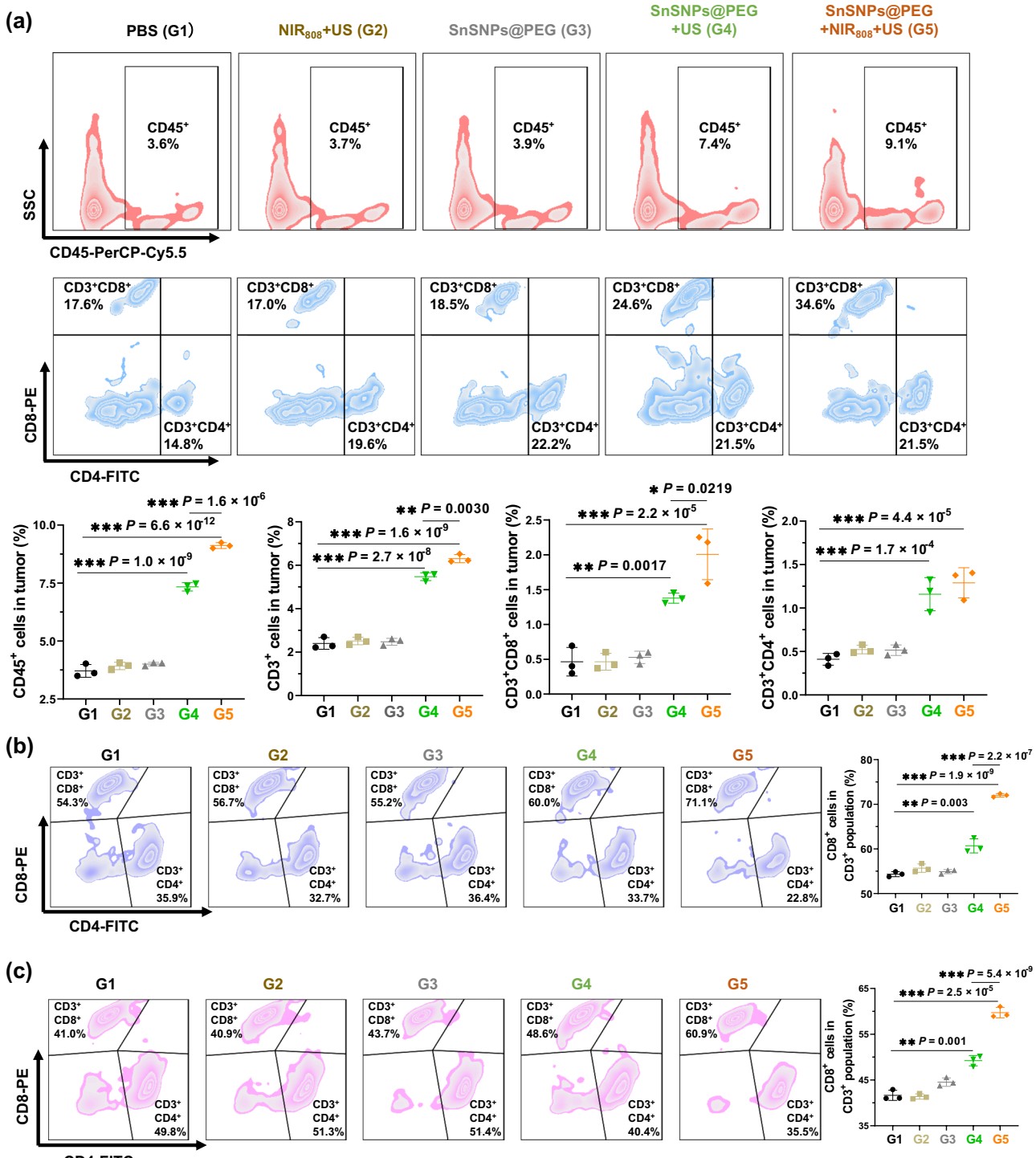

**Fig. 8 | Flow cytometry analysis of T lymphocytes in 4T1 tumor-bearing mice after treatment. a** Flow cytometry analysis of CD45⁺ immune cells, CD45⁺CD3⁺ lymphocytes, CD45⁺CD3⁺CD4⁺ T lymphocytes and CD45⁺CD3⁺CD8⁺ T lymphocytes in the tumor of 4T1 tumor-bearing mice after various treatments ($n = 3$). **b** Flow cytometry analysis of CD45⁺CD3⁺CD4⁺ T lymphocytes and CD45⁺CD3⁺CD8⁺ T lymphocytes in spleen of 4T1 tumor-bearing mice after various treatments ($n = 3$).

**c** Flow cytometry analysis of CD45⁺CD3⁺CD4⁺ T lymphocytes and CD45⁺CD3⁺CD8⁺ T lymphocytes in lymph node of 4T1 tumor-bearing mice after various treatments. Data are presented as mean ± SD ($n = 3$). Multiple comparisons among groups were performed by one-way ANOVA. *$P < 0.05$, **$P < 0.01$, ***$P < 0.001$. PBS, phosphate-buffered saline; US, ultrasound; NIR, near-infrared.

The development of a highly effective sonosensitizer is essential for solid tumor treatment. There are mainly two types of sonosensitizers, i.e., organic and inorganic sonosensitizers. Organic sonosensitizers derived from photosensitizers, such as photofrin II and phthalocyanine[39], typically have poor biological stability, and some phototoxicity reports have been noted[40]. Inorganic sonosensitizers, such as semiconducting TiO₂[41,42] and zinc oxide (ZnO)[43] have wide bandgaps (3.2 eV[18] and 3.3 eV[44], respectively) and rapid recombination of e⁻ and h⁺, that limit the yield of ROS, making them less effective for SDT. Remarkably, a narrow bandgap can facilitate the US-triggered generation of ROS as it requires less energy for electron excitation and separation of electron-hole pairs[45]. More specifically, regarding the US-

triggered generation of ROS by the sonosensitizers, under US exposure, the electrons in the valence band of the sonosensitizers adsorb energy and are excited to the conduction band, generating electron-hole pairs followed by energy release from the US-activated SnSNPs through a radiative recombination process. In the biological environment, surrounding $O_2$ and $H_2O$ molecules capture the released energy and the holes to generate $^1O_2$ and ·OH, respectively[23,46]. In our study, the $E_g$ of SnSNPs was measured to be 1.18 eV (Fig. 3a), which is consistent with previous reports[47,48]. Notably, the $E_g$ of SnSNPs was found to be narrower compared to other previously studied nano-sono-sensitizers, as summarized in Table S1. Specifically, compared to $BiVO_4$ ($E_g$: ~2.5 eV)[20], $TiH_{1.924}$ ($E_g$: ~2.7 eV)[30] and $TiO_2$ ($E_g$: ~3.2 eV)[49], the relatively narrow $E_g$ of SnSNPs can facilitate ROS generation by US irradiation, leading to better outcomes for tumor SDT. For example, $TiO_2$/C nanocomposites-mediated SDT only suppressed tumor growth by three times of treatment with an interval of two days, but this treatment regimen did not eradicate tumors[50]. Moreover, a single dose injection of $BiVO_4$ nanorods followed by two US irradiation treatments temporarily suppressed tumor growth but tumors continued to grow soon after the treatment[20]. In contrast, SnSNPs-mediate SDT efficiently suppressed tumor growth with a single treatment and successfully eradicated 4T1-tumors with two times combination treatment (US + NIR, Fig. 6), demonstrating a superior ROS generation ability and SDT efficacy of SnSNPs due to their narrow bandgap.

In addition to the strong SDT performance, SnSNPs exhibited mild photothermal effects when exposed to $NIR_{808}$ irradiation (Fig. S7). Photothermal therapy (PTT) has shown great promise for tumor treatment[51,52], and combining it with SDT has also been reported to enhance therapeutic outcomes[53]. However, conventional PTT typically requires high temperature (over 50 °C) for effective tumor treatment, which can cause damage to normal tissues by inducing the synthesis of heat shock protein[54], overproduction of ROS in cells[55], and triggering inflammatory responses[56,57]. Therefore, conventional PTT has limitations and side effects due to low selectivity and fast heat diffusion from tumor tissues to normal tissues[58]. In recent years, mild PTT (~45 °C) has emerged as a complementary therapeutic strategy with minimal side effects[59]. To take advantage of this strategy, we designed a denaturation-and-penetration approach that utilizes the mild photothermal effects of SnSNPs to enhance SDT and promote antitumor immunity. In contrast to PTT, SDT mediated by SnSNPs induces tumor cell death by increasing ROS levels. As a result, a sufficient uptake of the sonosensitizer by tumor cells and its accumulation at the tumor site are prerequisites for exerting desirable SDT effects. Despite the improved infiltration of SnSNPs@PEG into the tumor (Fig. 7b), a large proportion of tumor cells showed little uptake of the sonosensitizer, making it unlikely to kill all tumor cells with a one-time treatment. Our in vivo studies confirm that a one-time treatment effectively suppresses tumor growth (Fig. 6) and, remarkably, a two-time treatment successfully eradicated the tumor in mice without recurrence (Fig. 6). Similar therapeutic results using SDT have been reported in previous studies, and multiple treatments or combined therapies are often used to improve efficacy[60–62]. However, we believe that the therapeutic efficacy of our strategy based on SnSNPs can be further enhanced by modifying the tumor targeting and refining the material design.

The biocompatibility of inorganic materials for therapeutic applications is a major concern[12]. We evaluated the biosafety of the external stimuli (i.e., $NIR_{808}$ and US) by testing cell viability without SnSNPs@PEG in vitro (Fig. 4e, f), which showed no decrease in viability, indicating the safety of $NIR_{808}$ and US exposure without the sonosensitizer. Additionally, our in vivo assessment demonstrated the high biocompatibility of both $NIR_{808}$ and US (Figures S10, S14). It is worth mentioning that, despite the high biocompatibility of US in our tests and other reports[63], we used a relatively low-power-density US (0.3 W cm$^{-2}$)[64] in in vitro study to avoid interference of cell detachment to the culture plate. Although the low-power-density US may have limited the ultrasonic performance

of SnSNPs@PEG, SnSNPs@PEG still demonstrated remarkable ROS generation ability for cytotoxicity (Fig. 4e–g). Additionally, our in vitro (Figs. 4e, f, S11) and in vivo assessments (Figs. S10, S14) showed no side effects following SnSNPs@PEG treatment, consistent with the high biocompatibility of SnS demonstrated before[24]. Notably, the spleen weight in mice recovered to a normal state in the two-time treatment group. In contrast, the spleen weight in other groups was higher than normal (Fig. S15), suggesting that the tumor eradication may have decreased the spleen burden. This observation further confirmed the safety and therapeutic efficacy of SnSNPs@PEG-mediated therapy. A similar observation on the spleen weight has been reported before[65]. Given this tumor accumulation ability, this therapeutic strategy based on highly biocompatible SnSNPs@PEG is unlikely to cause undesired damage to normal tissues.

In summary, we have designed an innovative denaturation-and-penetration strategy and developed SnSNPs as a nano-sonosensitizer for enhanced SDT and antitumor immunity. The SnSNPs-based therapeutic strategy demonstrated excellent treatment efficacy in orthotopic mouse models of TNBC and HCC by noninvasively killing tumor cells through efficient ROS generation, boosting antitumor immunity, and increasing the infiltration of tumor-specific CTLs that further contributed to the eradication of tumor. Therefore, the denaturation-and-penetration strategy employing SnSNPs represents a promising nanoplatform for SDT and antitumor immunity for tumor treatment. In addition to SnSNPs, other nanomaterials and therapeutics with similar photothermal, sonodynamic, or other properties, such as chemodynamic therapy, may also take the advantage and implement the therapeutic strategy developed in this work.

## Methods
### Materials and chemicals
Tin monosulfide powder (99.5%) was purchased from Alfa Aesar. Propidium iodide (PI, 1.0 mg mL$^{-1}$ in water), 1,3-diphenylisobenzo-furan (DPBF, 97%), $N$-methylpyrrolidone (NMP, 99.7 + %), ethanol (99.8 + %), reduced glutathione (GSH, 98 + %), 5,5′-dithiobis-(2-nitrobenzoic acid) (DTNB, 98 + %), methylene blue (MB), 2′,7′-dichlorofluorescin diacetate (H$_2$DCF-DA, 97 + %), and acetic acid were purchased from Sigma-Aldrich. Sodium hydroxide was purchased from Macron Fine Chemicals. Calcein-AM (90 + %) was purchased from Biotium. 1,2-Distearoyl-$sn$-glycero-3-phosphoethanolamine-$N$-[methoxy(polyethylene glycol)] (DSPE-PEG, Mw: 5000 Da) were purchased from Laysan Bio. DSPP-PEG-Cy5 (MW: 5000 Da) was purchased from NANOCS. Roswell Park Memorial Institute 1640 (RPMI 1640), fetal bovine serum (FBS), L-glutamine, antibiotics (10,000 Units mL$^{-1}$ penicillin and 10,000 μg mL$^{-1}$ streptomycin) and Hoechst 33342 were purchased from Invitrogen. Trypsin-ethylenediaminetetraacetic acid (trypsin-EDTA) (0.25%) was purchased from Corning. Phosphate-buffered saline (PBS) was purchased from HyClone. AlamarBlue and paraformaldehyde solution (4% in PBS) were purchased from Fisher Scientific. Hydrochloric acid (HCl) and nitric acid (HNO$_3$), BDH Aristar® Plus for trace metal analysis, were purchased from The Lab Depot.

### Instruments and characterization
The size and morphology of SnSNPs were investigated by transmission electron microscopy (TEM, Tecnai G2 F20, FEI, Netherlands). The dynamic light scattering (DLS) size and zeta potential of SnSNPs@PEG were determined by a laser particle analyzer (ZetaPALS, Brookhaven Instruments). The diffuse reflectance spectrum of SnSNPs was measured by a UV−Vis−NIR spectrophotometer (Shimadzu UV-3600i Plus, Japan). The chemical composition of SnSNPs was determined by X-ray photoelectron spectroscopy (XPS, ESCALAB 250Xi, Thermo Scientific). The X-ray diffraction pattern of SnSNPs was obtained by a MiniFlex600 diffractometer with Cu Kα radiation (λ = 1.5418 Å). The functional

groups on the surface of SnSNPs and SnSNPs@PEG were characterized by Fourier transform infrared (FTIR) spectrometer (iS50 ABX, USA) in the range of 4000–400 cm$^{-1}$. Thermogravimetric analysis (TGA) was performed on a Synchronous DSC-TGA Thermal Analysis (SDT Q600) machine under nitrogen flow (50 mL min$^{-1}$) at a stable heating rate of 10 °C min$^{-1}$. The concentration of Sn in tumors was determined by the High Resolution (magnetic sector field)-inductively coupled plasma mass spectrometry (HR-ICP-MS) (Nu Instruments AttoM ES).

## Synthesis of SnSNPs

SnSNPs were prepared by a liquid-phase exfoliation strategy through direct probe sonication in NMP. Briefly, 100 mg of bulk SnS powder was dispersed in 20 mL of NMP solution and sonicated with an ultrasound probe under an ice water bath (ON/OFF = 10 s/5 s, 500 W, 40% power, 20 kHz, FB505, Fisher Scientific) for 20 h. Afterward, the solution was centrifuged at 1000 g/2900 rpm for 5 min, and the supernatant was collected and further centrifuged at 21,100 g/14,800 rpm for 15 min to obtain the SnSNPs. SnSNPs were washed three times with ethanol, resuspended in ethanol, and stored in a freezer (−20 °C) before use.

## Surface modification of SnSNPs with PEG

The surface of SnSNPs was modified by DSPE-PEG to enhance dispersity and colloidal stability in biological environment as well as its blood circulation time. To obtain PEG-coated SnSNPs, 10 mL of ethanol containing 2 mg of SnSNPs and 10 mg of DSPE-PEG was mixed and sonicated with an ultrasound probe (ON/OFF = 15 s/5 s, 500 W, 20% power, 20 kHz, FB505, Fisher Scientific) under an ice water bath for 30 min. Then, rotary evaporation was performed to vaporize ethanol. The dry film of PEG-coated SnSNPs (SnSNPs@PEG) on the glass flask was resuspended in Hypure water using a water bath sonication. Uncoated PEG was removed by centrifugation at 21,100 g/14,800 rpm for 10 min and washed three times with Hypure water. To obtain DSPE-PEG-Cy5-coated SnSNPs, 1 mg of SnSNPs in ethanol was mixed with 2 mg of DSPE-PEG-Cy5 (MW: 5,000 Da). The mixed solution was sonicated with an ultrasound probe (ON/OFF = 15 s/5 s, 500 W, 20% power, 20 kHz, FB505, Fisher Scientific) under ice water bath for 30 min. Then, ethanol was vaporized by rotary evaporation. Free DSPE-PEG-Cy5 was removed by centrifugation at 21,100 g/14,800 rpm for 10 min and washed three times with PBS. Purified SnSNPs@PEG-Cy5 were stored in a dark environment at 4 °C and used within a week.

## Photothermal performance of SnSNPs@PEG

The concentration-dependent photothermal performance of SnSNPs@PEG was evaluated by exposing 1 mL of cell culture medium containing various concentrations of SnSNPs@PEG (0, 50, 100, and 200 μg mL$^{-1}$) to an 808 nm NIR laser (BWF, B&W TEK) at a power density of 2.0 W cm$^{-2}$. In addition, the power density-dependent (0, 0.5, 1.0, 1.5, and 2.0 W cm$^{-2}$) photothermal performance of SnSNPs@PEG was measured by exposing 1 mL of cell culture medium containing 200 μg of SnSNPs@PEG to the 808 nm NIR laser for 10 min. The real-time temperature of the solutions was measured by an IR thermal camera (T100, 9 Hz, Fluke®). For multiple cycles of NIR irradiation, 1 mL of the cell culture medium containing 200 μg of SnSNPs@PEG was exposed to the 808 nm NIR laser for 10 min (ON) followed by naturally cooling to room temperature for 20 min (OFF). A total of five ON/OFF cycles were carried out to evaluate the photothermal stability of SnSNPs@PEG. The photothermal conversion efficiency (PTCE) of SnSNPs@PEG was calculated as described previously as follows Equation[66].

$$\eta = \frac{hA\,\Delta T_{\max} - Q_s}{I(1 - 10^{-A_{808}})}$$

where $h$ is the coefficient of heat transfer, $A$ is the container surface area, $\Delta T$ is the temperature change of the solution of SnSNPs@PEG, $I$ is the power density of the NIR laser, $A_{808}$ is the absorbance of the solution of SnSNPs@PEG at 808 nm, and $Q_s$ is the heat associated with the light absorbance of the solution.

## US-triggered generation of $^1O_2$ by SnSNPs

To evaluate the US-triggered generation of $^1O_2$, SnSNPs@PEG (200 μg mL$^{-1}$) dispersed in 1 mL of PBS was mixed with 1 mL of ethanol containing DPBF (80 μg mL$^{-1}$). After stirring in a dark environment for 5 min, the mixture was exposed to ultrasound irradiation (1 MHz, 1 W cm$^{-2}$, 50% duty cycle) (SoundCare Plus, 1 cm$^2$ probe, Roscoe Medical) for 5, 10, 15, 20, and 25 min. Groups of DPBF + US and DPBF + SnSNPs@PEG were carried out as controls. The resulting solution was measured by a UV−Vis−NIR spectrometer (350‑550 nm) and the absorbance at 410 nm showing the degradation of DPBF was measured to quantify the generation of $^1O_2$.

## US-triggered generation of ·OH by SnSNPs

The MB assay was performed to evaluate the US-triggered generation of ·OH. SnSNPs@PEG (100 μg mL$^{-1}$) dispersed in 1 mL of PBS was mixed with 1 mL of MB (5 μg mL$^{-1}$) with or without $H_2O_2$ (50 μM). Then, the mixture was irradiated by US (1 MHz, 2 W cm$^{-2}$, 50% duty cycle) for different periods of time (5, 10, 15, 20, 25, and 30 min). The resulting solution was measured by a UV−Vis−NIR spectrometer (500–800 nm), and the absorbance changes at 668 nm were recorded to assess the generation of ·OH. Groups of MB + SnSNPs@PEG, MB + $H_2O_2$, MB + SnSNPs@PEG + $H_2O_2$, MB + $H_2O_2$ + US, MB + US were carried out as controls.

## US-triggered GSH depletion of SnSNPs@PEG

GSH concentration was quantified by DTNB, an Ellmann probe. PBS solution containing SnSNPs (200 μg mL$^{-1}$) and GSH (30 μg mL$^{-1}$) were irradiated with US (1 MHz, 2 W cm$^{-2}$, 50% duty cycle) for different periods (5, 10, 15, 20, 30, and 40 min). The SnSNPs@PEG in the solution was removed by centrifugation (21,100 g/14,800 rpm × 10 min), and the supernatant (190 μL) was mixed with 10 μL of DMSO solution containing DTNB (0.5 mg mL$^{-1}$) and then measured by a UV−Vis−NIR spectrometer (350–500 nm). The absorbance at 412 nm of the solution was recorded to evaluate the consumption of GSH. Groups of GSH + SnSNPs@PEG, GSH + US, GSH + NIR$_{808}$ (1.0 W cm$^{-2}$) and GSH + SnSNPs@PEG + NIR$_{808}$ were carried out as controls.

## Cell culture

Mouse triple-negative breast cancer cell line 4T1 (CRL-2539) and the luciferase-expressing cell line 4T1-luc (CRL-2539-LUC2) were obtained from American Type Culture Collection (ATCC). The murine hepatocellular carcinoma (HCC) cell line RIL-175-luc was obtained from Prof. Dan G. Duda's lab at Massachusetts General Hospital, Boston, USA. These tumor cells were cultured in Dulbecco's Modified Eagle Medium (DMEM) supplemented with 10% FBS and 1% Penicillin-Streptomycin. Cells were incubated at 37 °C in a humidity-controlled incubator with 5% CO$_2$. The cells were harvested by trypsinization with trypsin-EDTA every two days.

## In vitro biocompatibility

The cell viability was examined by an AlamarBlue assay. 4T1 cells were seeded in 96-well plates at a density of $1 \times 10^4$ cells per well. After attachment overnight, the medium was removed and the cells were incubated in 150 μL of fresh medium containing different concentrations of SnSNPs@PEG (0, 25, 50, 100, 200, and 400 μg mL$^{-1}$) for 24 h or 48 h in an incubator. Afterward, the medium was removed, and the treated cells were washed two times using PBS and then incubated with fresh medium (100 μL) containing 10% of AlamarBlue reagent. After 30 min of incubation at 37 °C, the fluorescence emission intensity at 590 nm (excitation at 545 nm) showing the viability of cells in each condition was measured by

a plate reader (Tecan Infinite M200PRO). The fluorescence of blank medium (100 μL) with AlamarBlue reagent was set as control.

### Cellular uptake of SnSNPs@PEG-Cy5

4T1 cells were seeded in a 24-well plate at a density of $1 \times 10^5$ cells per well and cultured overnight for attachment. Then, the medium was removed and replaced with a fresh medium containing SnSNPs@PEG-Cy5 (200 μg mL$^{-1}$) for 2, 6, or 24 h at 37 °C in an incubator. After washing 3 times with PBS, the treated cells were fixed with 4% paraformaldehyde for 15 min at room temperature (21 °C) in a dark environment. Then, the fixed cells were washed three times using PBS with slight shaking. Afterward, the nuclei of the cells were stained with Hoechst 33342 (1 mL, 8 μg mL$^{-1}$) for 15 min in a dark environment followed by washing 3 times with PBS (1 mL). Finally, the stained cells were covered by a mounting medium and cover glass before being visualized under a confocal laser scanning microscope (FV1000, Olympus). The cellular uptake of SnSNPs@PEG was also observed directly with a microscope (XL core system, EVOS).

### In vitro SDT and combinational therapy

4T1 cells were seeded in a 24-well plate at a density of $1 \times 10^5$ cells per well. After attachment, the cells were incubated with a fresh medium containing SnSNPs@PEG (200 μg mL$^{-1}$) for 24 h. For SDT, the cells were treated with US (1 MHz, 0.3 W cm$^{-2}$, 50% duty cycle) for 3 min, rested for 5 min, and then treated with the US for another 3 min. For combinational therapy, cells were treated with NIR laser irradiation (808 nm, 1.0 W cm$^{-2}$) for 5 min, rested for 10 min, and then treated with US. Groups of PBS, US, NIR, US + NIR, SnSNPs@PEG, and SnSNPs@PEG + NIR were carried out as controls. The viability of cells was evaluated by the AlamarBlue assay at 24 h post-treatment. Cell morphology was directly observed by a microscope (XL core system, EVOS). The cell colony was stained with crystal violet (0.005%), and the colony area was analyzed using the ImageJ software. Also, Calcein-AM/PI co-staining assay was performed to observe the cell death. Specifically, 4T1 cells were seeded in a 24-well plate at a density of $1 \times 10^5$ cells per well and received SDT and combination therapy as described above. Cells were stained with Calcein-AM (4 μg/mL) and PI (10 μg/mL) for 30 min at 37 °C and washed three times with PBS. Finally, Calcein-AM/PI co-stained cells were observed under a fluorescence microscope (Axio Vert A1, ZESIS).

### Intracellular ROS detection

4T1 cells were seeded in a 96-well plate at a density of $1 \times 10^4$ cells per well. After attachment, the cells were cultured with fresh medium containing SnSNPs@PEG (200 μg mL$^{-1}$) and incubated for 24 h. The procedure of cell treatment was the same as that described in vitro combinational therapy followed by incubation at 37 °C in an incubator for 8 h. Afterward, cells were washed 3 times with PBS followed by incubating with 100 μL of PBS containing H$_2$DCF-DA (50 μM) at 37 °C for 30 min. Afterward, the cells were fixed with 4% paraformaldehyde for 15 min at room temperature. Subsequently, the cells were stained with Hoechst 33342 (6 μg mL$^{-1}$) for 15 min and washed three times with PBS. Finally, the cells were visualized under a fluorescence microscope (Axio Vert A1, ZESIS).

### In vitro matrix denaturation assay and diffusion measurement

To test the ability of SnSNPs to denature collagen matrix under NIR irradiation, we prepared tumor stroma collagen-mimicking gel as follows: 100 μL of collagen I (5 mg mL$^{-1}$) was mixed with 15 μL of PBS (10×), 3 μL of NaOH (4 mg mL$^{-1}$) and 50 μL of RPMI 1640 medium (containing 10% FBS) on ice. Thirty μL of the mixture was added from a reservoir to fill pre-cooled channels (μ-Slide VI 0.4, ibidi) followed by incubation at 37 °C overnight to form intact collagen gel. Then, 30 μl of PBS containing SnSNPs@PEG (200 μg/mL) was added to both sides of the channel. One side was exposed to NIR irradiation (1 W cm$^{-2}$, 808 nm) for 10 min, and the temperature was recorded. After NIR irradiation, the channel was imaged using a microscope (XL core system, EVOS) for a head-to-head comparison of SnSNPs@PEG penetration in the channel.

### Animals

The animal protocol was approved by the Institutional Animal Care and Use Committees at Brigham and Women's Hospital, Harvard Medical School. All in vivo studies were performed following National Institutes of Health animal care guidelines. Six-week-old female BALB/cJ mice and C57BL/6J mice were purchased from the Jackson Laboratory. All animals were housed in individually ventilated cages with 12-hour alternate light and dark cycles and at controlled ambient temperature (68-79F) with humidity between 30% and 70%. Orthotopic breast tumors were established by implanting $1 \times 10^7$ 4T1-luc cells (in 60 μL of PBS) into the mice left fourth mammary fat pad. Orthotopic HCC was established by implanting $2 \times 10^6$ RIL175-luc cells (suspended in 25 μL of 50 v/v% Matrigel and PBS) into the lower margin of the liver. The tumor growth was monitored by the in vivo imaging system (IVIS, Lumina LT Series III, PerkinElmer) and tumor volume measurement (Width$^2$ × Length/2) with a digital measurement caliper. The maximum allowed tumor size is 2,000 mm$^3$ and no tumor-bearing mouse exceeded the limit.

### Biodistribution and tumor penetration

The biodistribution of SnSNPs@PEG-Cy5 was evaluated on 4T1 tumor-bearing mice. SnSNPs@PEG-Cy5 (10 mg kg$^{-1}$) was administrated to mice via i.v. injection. The fluorescence was observed at different time points (2 h, 6 h, 12 h, and 24 h) by an imaging system (PXi 4 Touch, Syngene). Ex vivo biodistribution was also performed 24 h after injection. Free DSPE-PEG-Cy5 dye was also injected into 4T1-bearing mice as a control.

To evaluate the tumor penetration, SnSNPs@PEG-Cy5 was first intravenously injected into 4T1 tumor-bearing mice. Twelve hours later, the tumor site was exposed to NIR irradiation for 10 min. Fifteen min later, tumors were excised and embedded with Tissue-Tek O.C.T (Sakura Finetek, USA), frozen with dry ice, and sectioned with a thickness of 15 μm (Leica cryostat CM1950). Tumor without NIR irradiation was set as control. Tumor sections were stained with Hoechst 33342 (1 mL, 8 μg mL$^{-1}$) for 15 min in a dark environment followed by washing with PBS 3 times (1 mL). Finally, the tumor tissue slices were imaged by a fluorescence microscope (Axio Vert A1, ZESIS).

To further study the enhanced tumor penetration of SnSNPs@PEG after NIR irradiation and analyze the concentration of Sn, we conducted ICP-MS analysis to quantify the amount of Sn in the tumor. The excised tumor samples were weighted and placed into 7 mL glass vials. Then, the samples were soaked in 5 mL of aqua regia, which was prepared by mixing trace-metal grade HNO$_3$ and HCl at a 1:3 ratio. The samples were digested overnight at 90 °C to ensure complete evaporation of aqua regia. After cooling to room temperature, 5 mL of 2% trace-metal grade HNO$_3$ were added to each digestion glass vial followed by sonication for 60 s to dissolve metals. Finally, samples were purified using filters (pore size: 0.22 μm, Millipore) and diluted with 2% trace-metal grade HNO$_3$ before conducting ICP-MS analysis.

### In vivo enhanced SDT

Mice bearing 4T1-luc tumor were randomly divided by six groups: (1) control, (2) SnSNPs@PEG, (3) NIR + US, (4) SnSNPs@PEG + US, (5) SnSNPs@PEG + NIR + US, (6) SnSNPs@PEG + NIR + US × 2 (N = 5). For the control group, mice received i.v. injection of 100 μL of PBS; for the SnSNPs@PEG group, mice received i.v. injection of SnSNPs@PEG (10 mg kg$^{-1}$) in 100 μL of PBS; for NIR + US group, mice firstly received i.v. injection of 100 μL of PBS. Twelve hours later, the tumor site was irradiated by NIR$_{808}$ (1 W cm$^{-2}$) for 10 min, maintaining the temperature at the tumor site is about 44 °C. Five min later, the tumor site was exposed to US irradiation (2 × 5 min with an interval of 5 min, 1 MHz,

2 W cm$^{-2}$, 50% duty cycle); for SnSNPs@PEG + US group, mice received *i.v.* injection of SnSNPs@PEG (10 mg kg$^{-1}$) in 100 μL of PBS. Twelve hours later, the tumor site was exposed to US irradiation (2 × 5 min with an interval of 5 min, 1 MHz, 2 W cm$^{-2}$, 50% duty cycle); for the SnSNPs@PEG + NIR$_{808}$ + US group, mice firstly received *i.v.* injection of SnSNPs@PEG (10 mg kg$^{-1}$) in 100 μL of PBS. Twelve hours later, the tumor site was exposed to NIR$_{808}$ + US treatment; for the SnSNPs@PEG + NIR$_{808}$ + US group × 2, mice received two times of treatment with an interval of three days. The body weights and tumor sizes of the mice were measured every two days; tumor biolumines-cence was measured on day 0, 2, 6, 10, and 16. Then, the mice were sacrificed, and the tumor and spleen tissues were harvested for pho-tography and weighting.

For histological analysis, hematoxylin and eosin (H&E) staining, Masson's staining, and terminal deoxynucleotidyl transferase-mediated dUTP-biotin nick end labeling (TUNEL) staining were per-formed. Tumors were collected 24 h after different treatments and then formalin-fixed, paraffin-embedded and sectioned for staining. In addition, for in situ ROS staining, right after treatments, tumor tissues were excised and embedded with Tissue-Tek O.C.T (Sakura Finetek) and frozen, followed by being sectioned at a thickness of 15 μm. The tumor tissue slices were stained by H$_2$DCF-DA (20 μM) for 30 minutes and washed three times using PBS. Then, the tumor sections were stained by Hoechst 33342 and washed 3 times with PBS. Finally, the tumor tissue slices were imaged by a fluorescence microscope to evaluate the ROS generation efficacy of different treatments.

To evaluate the deep-tissue cancer treatment efficacy of enhanced SDT using SnSNPs@PEG, we established an orthotopic HCC mouse model, which was confirmed using IVIS. The tumor-bearing mice were randomly assigned to one of the four groups: (1) control, (2) SnSNPs@PEG, (3) SnSNPs@PEG + US, (4) SnSNPs@PEG + NIR + US ($N = 4$). The mice received treatment similar to the 4T1-luc tumor-bearing mice with the same injection dose and SDT procedures. Bio-luminescence images and intensity of tumor-bearing mice were cap-tured after tumor inoculation and during treatment. After four imaging sessions, the livers of tumor-bearing mice were harvested to visually assess the tumor size and evaluate treatment efficacy.

### Enhanced SDT induces anti-tumor immunity

To evaluate whether the enhanced SDT can induce anti-tumor immunity and cytotoxic T lymphocyte infiltration into tumors, we performed flow cytometry to analyze the percentage of T lympho-cytes in tumors, spleens, and lymph nodes in 4T1-tumor-bearing mice. Forty-eight hours after treatment, tumors, spleens and lymph nodes were excised and washed using PBS. Samples were cut into small pieces and filtered through cell strainers (70-μm filter for tumor, 40-μm filter for spleen and lymph node) to obtain single-cell suspension. After removing red blood cells, $1 \times 10^6$ cells were stained with fixable viability dye (Cat# 65-0866-14, eBioscience, 1.0 μg/mL) and incubated with anti-CD16/32 antibody for 30 min (Cat# 130-092-575, Miltenyi Biotec, 0.5 μg/mL) to block cell surface Fc-receptor. Afterward, the cells were stained with a cocktail solution containing anti-CD45-PerCP-Cy5.5 (Cat# 109828, Biolegend, 2.0 μg/mL), anti-CD3-APC (Cat# 100236, Biolegend, 2.5 μg/mL), anti-CD4-FITC (Cat# 100406, Biolegend, 1.5 μg/mL), anti-CD8a-PE (Cat# 100708, Biole-gend, 1.5 μg/mL) antibodies. Stained cells were washed two times with PBS (300 g/1560 rpm × 5 min) to remove unbounded antibody-conjugated fluorescent dyes. The concentration of antibodies that were used for flow cytometry analysis is summarized in Table S2. Finally, cells were analyzed by flow cytometry (LSRFortessa, BD) equipped with 405, 488, 561, and 640 nm lasers. All staining pro-cesses were performed in a dark environment at 4 °C. CD3$^+$, CD4$^+$, and CD8$^+$ T lymphocytes were identified as CD45$^+$CD3$^+$, CD45$^+$CD3$^+$CD4$^+$, and CD45$^+$ CD3$^+$ CD8$^+$ cells, respectively. For HCC

mouse models, the T lymphocytes in HCC were also analyzed using the same staining and flow cytometry procedures.

### Reporting summary

Further information on research design is available in the Nature Portfolio Reporting Summary linked to this article.

## Data availability

All study data are available within this manuscript and the associated Supplementary Information. Other data are available from the corre-sponding author upon reasonable request. Source data are provided with this paper.

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

## Acknowledgements

This work was supported by US METAvivor Early Career Investigator Award (No. 2018A020560, W.T.), Harvard/Brigham Health & Technology Innovation Fund (No. 2023A004452; W.T.), Department Basic Scientist Grant (No. 2420 BPA075, W.T.), Gillian Reny Stepping Strong Center for Trauma Innovation Breakthrough Innovator Award (No. 113548, W.T.), Nanotechnology Foundation (No. 2022A002721, W.T.), Farokhzad Family Distinguished Chair Foundation (No. 018129, W.T.), National Natural Science Foundation of China (No. 82122076, N.K.), and National Natural Science Foundation of China (No. 81730108 and 81973635, T.X.). W.T. also acknowledges the support from American Heart Association (AHA) Transformational Project Award (No. 23TPA1072337), AHA Collaborative Sciences Award (No. 2018A004190), AHA's Second Century Early Faculty Independence Award (No. 23SCEFIA1151841), American Lung Association (ALA) Cancer Discovery Award (No. LCD1034625), ALA Courtney Cox Cole Lung Cancer Research Award (No. 2022A017206), Novo Nordisk Validation Award (No. 2023A009607), and the Khoury Innovation Award (No. 2020A003219).

## Author contributions

Y.L., W.C., and W.T. conceived and designed the study. Y.L., W.C., Y.K., X.Z., Z.Z., C.L., S.C., X.H., H.-J.L., S.K., N.K., X.J., T.X. performed the experiments and analyzed the data. N.K., T.X. and W.T. supervised the project. Y.L. and W.C. drafted the manuscript. All authors participated in the interpretation of the data and production of the final manuscript.

## Competing interests

The authors declare no competing interests.
