## [Peer Review File · Nature Communications]

Nanosensitizer-mediated augmentation of sonodynamic therapy efficacy and antitumor immunityREVIEWER COMMENTS

Reviewer #1 (Remarks to the Author):

In this study, the authors employ tin monosulfide nanodots (SnSNDs) as nano-sonosensitizers and photothermal agents to overcome the stromal barrier for the treatment of triple-negative breast cancer (TNBC). However, utilizing this strategy that mild photothermal effects to enhance the penetration of nanomaterials as well as different types of cancer therapies has been widely studied. The novelty of this study may be limited. Moreover, there are some major issues should be addressed:

1. Deep tissue penetration capability (on the order of centimeters) is the main advantage of ultrasound. The author should further investigate the therapeutic outcomes of SnSNDs on deep-sited tumor (e.g., osteosarcoma or orthotopic liver tumor) rather than subcutaneous tumor.
2. Do the authors notice the cavitation effects of SnSNDs during the ultrasound irradiation? This process would also generate large amounts of ROS.
3. The diameter of SnSNDs is 15 nm, the expression of SnS nanodots is wrong. Besides, the GSH depletion mainly attribute to the generated ROS rather than SnSNDs themselves. Please correct these expressions in the manuscript.
4. The cells in Fig. 3i seems not distinct. Please add DAPI staining in this data. In addition, the fluorescent imaging results in Fig. 6b is also unobvious. Utilizing ICP-OES to determine the Sn level inside of tumor would be an exact method to evaluate the tumor penetration of SnSNDs.
5. Do the authors compare the sonosensitization effects of SnSNDs with previous reported sonosensitizers under the same condition?

Reviewer #2 (Remarks to the Author):

This is an insightful manuscript reporting the use of a novel sonosensitizer, Tin monosulfide nanodots (SnSNDs), with a well-designed denaturation-and-penetration strategy to achieve enhanced sonodynamic therapy and antitumor immunity for the treatment of triple-negative breast tumors by overcoming the stromal barrier. Sonodynamic therapy is a potential cancer treatment modality, and this manuscript reported a highly efficient SnSNDs sonosensitizer and, more importantly, a useful strategy that can enhance the SDT effects as well as antitumor immunity. With the denaturation-and-penetration strategy, the authors further showed significantly enhanced penetration of SnSNDs and immune cells into the tumor, which led to superior antitumor effects. This strategy can also be helpful to other biomaterials with similar properties to overcome the stromal barrier, which is a critical issue for the treatment of multiple solid tumors. The design of experiments is dedicated to show the uniqueness and advantages of their concepts, and the authors have presented clear and robust data for a straightforward demonstration of their strategy. The Introduction is well-organized, and the discussion is insightful and informative. Overall, this reviewer would like to recommend the publication of this manuscript in Nature Communications after addressing the following comments.

1. One of the biggest questions when comes to inorganic nanoparticle is their biocompatibility and safety to be used systemically. The author has presented thorough characterization of SnSNDs in the manuscript, and is sufficient to demonstrate their compatibility, safety and their stability. The PEG coating on SnSNDs are an advantage to improve in vivo systemic administration. Although the synthesized SnSNDs showed good dispersibility after PEG coating, it would be beneficial to also include the zeta potential data for better characterization and demonstration of the stability and dispersibility of SnSNDs. Zeta potential is important as a near positive charged particle would be deemed less compatible in vivo due to their non-specific binding to negative-charged cell membranes.
2. Please verify the following inconsistency. In Figure 2h, the y-axis is labeled as absorbance at 668

nm, but in the Method section, the wavelength is stated as 654 nm. While the absorbance peak of methylene blue may vary based on the experimental conditions, the inconsistency between the two values needs to be corrected.

3. Line 132-133, Figure 2b is solely for the DPBF assay, while Figure 2e is for the MB assay, they should both be cited in line 133.

4. In Figure 3d, I would recommend the authors include more description regarding the parameters in the study design, such as wavelength of NIR, cell line information and US parameters. This additional information would provide readers with a better understanding of the experimental design.

5. The authors used low-intensity US in their in vitro assays, and although they have discussed the intensity concern in the Discussion section, it would be beneficial to mention it at the beginning of the corresponding description in the Results section to provide a clearer understanding for readers.

6. In the biodistribution study, the authors noted that the most prominent accumulation of SnSNDs at the tumor site occurred at 12 h after injection. It would be helpful if the authors could provide an explanation for why they chose to wait 24 hours after i.v. injection before irradiation and treatment in their experiments.?

7. The Discussion section is comprehensive and provides sufficient information in addition to the results. In the Discussion section, Line 400, it should not be low-intensity US as the authors used the same power of in-tube tests for in vivo studies.

8. There are a few minor points that could be improved in the manuscript. Firstly, the labeling color used for discriminating different groups could be made more consistent in Figure 3, Figure 6, and Figure 7. Additionally, the wavelength of NIR should be added to several descriptions where it is currently missing. Finally, there is a typo in Line 285, where "collagen" is misspelled.

RESPONSE TO REVIEWER COMMENTS

Reviewer #1 (Remarks to the Author):

In this study, the authors employ tin monosulfide nanodots (SnSNDs) as nano-sonosensitizers and photothermal agents to overcome the stromal barrier for the treatment of triple-negative breast cancer (TNBC). However, utilizing this strategy that mild photothermal effects to enhance the penetration of nanomaterials as well as different types of cancer therapies has been widely studied. The novelty of this study may be limited. Moreover, there are some major issues should be addressed:

Reply: Thank you very much for your thoughtful comment and valuable suggestions. We are truly grateful for your interest in our work. We would like to share that, to the best of our knowledge, this work represents the first report of SnSNPs as an innovative sonosensitizer with a narrow bandgap, which is particularly beneficial for ultrasound-mediated sonodynamic therapy for tumors. In addition to the nanoparticle-based sonosensitizer, we have developed an innovative “denaturation-and-penetration” strategy that harnesses the mild photothermal effects of SnSNPs to enhance penetration into tumors via the denaturation of tumor collagen. While it’s true that the reviewer points out that some studies have explored combinational photothermal and SDT for tumor therapy, it's essential to note that conventional PTT often requires high temperature, which can potentially damage normal tissues. Additionally, the enhanced tumor penetration, as investigated in our strategy, has not been previously explored. Therefore, we believe that our approach provides a unique and innovative solution. Our primary goal was to develop a strategy that facilitates in situ intratumoral penetration, not only of SnSNPs but also of cytotoxic lymphocytes, thus enabling more effective treatment of desmoplastic tumors. Through our approach, we achieved a robust therapeutic effect, showing significant promise for future advancements in this field. Once again, we sincerely appreciate your interest and feedback on our work. The point-by-point response is shown below.

1. Deep tissue penetration capability (on the order of centimeters) is the main advantage of ultrasound. The author should further investigate the therapeutic outcomes of SnSNDs on deep-sited tumor (e.g., osteosarcoma or orthotopic liver tumor) rather than subcutaneous tumor.

Reply: We appreciate the reviewer’s suggestion. We agree that the advantage of ultrasound can be further highlighted in deep-tissue tumor therapy. We therefore performed another therapeutic assay on orthotopic hepatocellular carcinoma (HCC), and the strategy based on SnSNPs successfully suppressed tumor growth. Additionally, we performed a flow cytometry analysis of immune cells within the tumor. We find that the SnSNPs-based therapy enhanced the immune response, leading to an increase in tumor-infiltrating cytotoxic T lymphocytes (CTLs) in the tumor. The corresponding data are presented in **Figure S16, S21, S22** as below.

We have included the new findings in the “**In vivo enhanced SDT of SnSNPs@PEG**” section of the Results section, as shown below.

“To assess the deep tissue penetration capability of SDT in enhanced tumor therapy, we developed an orthotopic HCC mouse model and evaluated the therapeutic effects. Our findings demonstrated significant anti-tumor efficacy of the SnSNPs@PEG-mediated therapeutic strategy (Figure S16a). Treatment with SnSNPs@PEG + US slightly slowed tumor progression, while SnSNPs@PEG + NIR + US treatment showed enhanced therapeutic effects, as evidenced by bioluminescence imaging (Figure S16b, S16c) and excised livers with tumors (Figure S16d). Similarly, there was no significant change in mice body weight during the treatment (Figure S16e). These results demonstrated that this SnSNPs-based therapeutic strategy is effective for deep-tissue tumor treatment.”

Figure S16. Antitumor efficacy of SnSNPs@PEG-mediated SDT in an orthotopic RIL-175-HCC mouse model. a) Experimental timeline for establishing the RIL-175-HCC mouse model and SnSNPs@PEG-mediated treatment; b) Bioluminescence images of orthotopic RIL-175-HCC-bearing mice before, during and after various treatments, including PBS (G1), SnSNPs@PEG (G2), SnSNPs@PEG + US (G3), and SnSNPs@PEG + NIR₈₀₈ + US (G4). The injection volume is 100 μ L and the dose of SnSNPs@PEG is 10 mg kg⁻¹; c) Analysis of bioluminescence (counts) of orthotopic RIL-175-HCC-bearing mice before, during and after various treatments; d) Photograph of excised livers with RIL-175-HCC after various treatments (Day 22). Tumor areas are highlighted by circles; e) Time-dependent body weight of RIL-175-HCC-bearing mice after receiving various treatments.

Figure S21. Flow cytometry analysis of (a) CD45⁺ cells, (b)(c) CD45⁺CD3⁺ lymphocytes, CD45⁺CD3⁺CD4⁺ T lymphocytes and CD45⁺CD3⁺CD8⁺ T lymphocytes in RIL-175-HCC after different treatments (n = 3). Statistical analysis among groups was performed using one-way ANOVA test. * P < 0.05, ** P < 0.01, *** P < 0.001. Groups are as follows: control (G1), SnSNDs@PEG (G2), SnSNDs@PEG + US (G3), SnSNDs@PEG + NIR₈₀₈ + US (G4).

Figure S22. Comparison of the levels of (a) CD45⁺ immune cells, (b) CD45⁺CD3⁺ T cells, (c) CD45⁺CD3⁺CD4⁺ T cells, and (d) CD45⁺CD3⁺CD8⁺ T cells in RIL-175-HCC-bearing mice after various treatments (n = 3). Statistical analysis among groups was performed using one-way ANOVA test. * P < 0.05, ** P < 0.01, *** P < 0.001. Groups are as follows: control (G1), SnSNDs@PEG (G2), SnSNDs@PEG + US (G3), SnSNDs@PEG + NIR + US (G4).

2. Do the authors notice the cavitation effects of SnSNDs during the ultrasound irradiation? This process would also generate large amounts of ROS.

Reply: Thank you for your valuable comment, and we genuinely appreciate your input regarding the significance of studying ROS generation in relation to the cavitation effects. The major strength of our study lies in the synthesis of innovative nanoparticle-based sonosensitizers and the "denaturation-and-penetration" strategy, which effectively harnesses the mild photothermal and sonodynamic effects of SnSNPs to enhance penetration into tumors and antitumor immunity. We believe that the mechanism for ROS generation is an important aspect, and therefore, we used various molecular probes to confirm the generation of ROS, including ¹O₂ and ·OH, from the US-triggered SnSNPs. This mechanism has also been widely investigated in previous studies. Moreover, we have confirmed the remarkable therapeutic effect through the enhanced sonodynamic therapy and "denaturation-and-penetration" strategy.

Indeed, the cavitation effects of SnSNDs during the ultrasound activation may be important for producing ROS, and we believe this area is worthy of future investigation. Again, we sincerely appreciate your comment on the potential cavitation effect, which warrants future systematic investigation.

3. The diameter of SnSNDs is 15 nm, the expression of SnS nanodots is wrong. Besides, the GSH depletion mainly attribute to the generated ROS rather than SnSNDs themselves. Please correct these expressions in the manuscript.

Reply: Thank you for your comment. We have revised our expression and described the material as SnS nanoparticles (SnSNPs). We have revised the expression throughout the manuscript. Regarding GSH depletion, we agree with the reviewer that the material itself cannot directly deplete GSH. It is the h⁺ generated by US-irradiated SnSNPs that consumed GSH. We have corrected our expression throughout the manuscript to avoid any misunderstanding.

4. The cells in Fig. 3i seems not distinct. Please add DAPI staining in this data. In addition, the fluorescent imaging results in Fig. 6b is also unobvious. Utilizing ICP-OES to determine the Sn level inside of tumor would be an exact method to evaluate the tumor penetration of SnSNDs.

Reply: We appreciate the reviewer's comment. As suggested, we have re-conducted the assay and included the nuclei staining in the new study. Please see below the new **Figure 3i** and **3j**. The new data still support the conclusion that SnSNPs@PEG triggered by the US could efficiently generate ROS in 4T1 cells.

Figure 3. (i) Fluorescence microscope images showing the intracellular ROS level detected by H₂DCF-DA probe (ROS, green). The cell nuclei were stained with Hoechst. (j) Analysis of the intensity of green fluorescence showing the relative ROS level. Data are presented as mean \pm SD. Statistical analysis between the two groups was performed by student's *t*-test. * $P < 0.05$, ** $P < 0.01$, *** $P < 0.001$.

In addition, to better determine the Sn level inside of tumor, we have measured the concentration of Sn in tumor tissues using ICP-MS after various treatments. The results (shown below) demonstrated an increased amount of Sn at the tumor site after NIR irradiation, further confirming the efficacy of our "denaturation-and-penetration" strategy based on SnSNPs. We have included this data in the **Figure 6c** of the revised manuscript and the statement in the "**Denaturation of tumor collagen and enhanced intra-tumoral penetration of SnSNPs@PEG**" section in the Result section.

It now reads:

"The results from the analysis using inductively coupled plasma-mass spectrometry (ICP-MS) also showed a significantly higher concentration of Sn at the tumor site in the SnSNPs@PEG + NIR₈₀₈ group compared to the group without NIR treatment (**Figure 6c**)."

Figure 6. (c). ICP-MS analysis of amount of Sn in 4T1-tumors with or without NIR₈₀₈ irradiation. Data are presented as mean \pm SD. Statistical analysis was performed by one-way ANOVA (n = 4). * $P < 0.05$.

5. Do the authors compare the sonosensitization effects of SnSNDs with previous reported sonosensitizers under the same condition?

Reply: We appreciate the reviewer's comment. The band gap of sonosensitizers is one of the most crucial factors in determining the electron and hole pair separation efficiency and sonodynamic efficacy and can be quantitatively compared. As per suggested by the reviewer, we have compiled the band gap values of representative nanoparticle-based sonosensitizers in Table S1 of the revised supporting information for comparison. Remarkably, our study revealed that the band gap of SnSNPs obtained is much narrower compared to those of representative nano-sonosensitizers. This finding highlights the potential advantages of SnSNPs as sonosensitizers. Therefore, to address this important observation, we have added a discussion of bandgap in the Discussion section below:

"Notably, the E_g of SnSNPs was found to be narrower compared to other previously studied nano-sonosensitizers, as summarized in Table S1. Specifically, compared to BiVO_4 (E_g : ~ 2.5 eV), $\text{TiH}_{1.924}$ (E_g : ~ 2.7 eV) and TiO_2 (E_g : ~ 3.2 eV), the relatively narrow E_g of SnSNPs can facilitate ROS generation by US irradiation, leading to better outcomes for tumor SDT."

The table has been included in **Table S1** of the supplementary information as shown below:

Table S1. Comparison of the bandgap of SnSNPs with other nano-sonosensitizers.

Sonosensitizer	Structure	Reported bandgap	Reference
SnS	Nanoparticle	1.18 eV	This work
Bi@BiO_{2-x}@Bi₂S₃	Nanoparticle	1.43 eV	1
α-Fe₂O₃@Pt	Heterostructure particle	1.83 eV	2
WO_x	Nanobelt	2.11 eV	3
Ti(Oi-Pr)₄@Ag	Metal-organic framework	2.11 eV	4
Sn	Nanosheet	2.3 eV	5
BiVO₄	Nanorod	2.5 eV	6
Vanadium carbide	Carbon dot	2.57 eV	7
Sodium molybdenum bronze	Nanoparticle	2.7 eV	8
TiH_{1.924}	Nanodot	2.7 eV	9
TiO₂	Nanoparticle	3.2 eV	10

Reference

1. Song, K., *et al.* Biodegradable Bismuth-Based Nano-Heterojunction for Enhanced Sonodynamic Oncotherapy through Charge Separation Engineering. *Adv. Healthc. Mater.* **11**, e2102503 (2022).
2. Zhang, T., *et al.* α -Fe₂O₃@Pt heterostructure particles to enable sonodynamic therapy with self-supplied O₂ and imaging-guidance. *J. Nanobiotechnology* **19**, 358 (2021).
3. Zhou, Y., *et al.* Oxygen-Deficient Tungsten Oxide (WO_x) Nanobelts with pH-Sensitive Degradation for Enhanced Sonodynamic Therapy of Cancer. *ACS Nano* **16**, 17242-17256 (2022).
4. Meng, X., *et al.* Ag-Doped Metal-Organic Frameworks' Heterostructure for Sonodynamic Therapy of Deep-Seated Cancer and Bacterial Infection. *ACS Nano* **17**, 1174-1186 (2023).
5. Chen, W., *et al.* Stanene-Based Nanosheets for β -Elemene Delivery and Ultrasound-Mediated Combination Cancer Therapy. *Angew. Chem. Int. Ed.* **60**, 7155-7164 (2021).
6. Yang, Z., *et al.* Conferring BiVO₄ Nanorods with Oxygen Vacancies to Realize Enhanced Sonodynamic Cancer Therapy. *Angew. Chem. Int. Ed.* **61**, e202209484 (2022).
7. Wang, H., *et al.* A MXene-derived redox homeostasis regulator perturbs the Nrf2 antioxidant program for reinforced sonodynamic therapy. *Chem. Sci.* **13**, 6704-6714 (2022).
8. He, X., *et al.* NIR-II photo-amplified sonodynamic therapy using sodium molybdenum bronze nanoplatform against subcutaneous Staphylococcus aureus infection. *Adv. Funct. Mater.* **32**, 2203964 (2022).
9. Gong, F., *et al.* Preparation of TiH_{1.924} nanodots by liquid-phase exfoliation for enhanced sonodynamic cancer therapy. *Nat. Commun.* **11**, 3712 (2020).
10. George, S., *et al.* Role of Fe doping in tuning the band gap of TiO₂ for the photo-oxidation-induced cytotoxicity paradigm. *J. Am. Chem. Soc.* **133**, 11270-11278 (2011).

Reviewer #2 (Remarks to the Author):

This is an insightful manuscript reporting the use of a novel sonosensitizer, Tin monosulfide nanodots (SnSNDs), with a well-designed denaturation-and-penetration strategy to achieve enhanced sonodynamic therapy and antitumor immunity for the treatment of triple-negative breast tumors by overcoming the stromal barrier. Sonodynamic therapy is a potential cancer treatment modality, and this manuscript reported a highly efficient SnSNDs sonosensitizer and, more importantly, a useful strategy that can enhance the SDT effects as well as antitumor immunity. With the denaturation-and-penetration strategy, the authors further showed significantly enhanced penetration of SnSNDs and immune cells into the tumor, which led to superior antitumor effects. This strategy can also be helpful to other biomaterials with similar properties to overcome the stromal barrier, which is a critical issue for the treatment of multiple solid tumors. The design of experiments is dedicated to show the uniqueness and advantages of their concepts, and the authors have presented clear and robust data for a straightforward demonstration of their strategy. The Introduction is well-organized, and the discussion is insightful and informative. Overall, this reviewer would like to recommend the publication of this manuscript in Nature Communications after addressing the following comments.

Reply: We sincerely appreciate the reviewer's concise summary of our manuscript. We are also grateful for the positive acknowledgment of the clarity and robustness of our data, which effectively demonstrate the outcomes of our well-designed study. Additionally, we are delighted to know that both the introduction and the discussion have been perceived as insightful and informative. Below, we present our point-by-point responses to the reviewer's valuable comments.

1. One of the biggest questions when comes to inorganic nanoparticle is their biocompatibility and safety to be used systemically. The author has presented thorough characterization of SnSNDs in the manuscript, and is sufficient to demonstrate their compatibility, safety and their stability. The PEG coating on SnSNDs are an advantage to improve in vivo systemic administration. Although the synthesized SnSNDs showed good dispersibility after PEG coating, it would be beneficial to also include the zeta potential data for better characterization and demonstration of the stability and dispersibility of SnSNDs. Zeta potential is important as a near positive charged particle would be deemed less compatible in vivo due to their non-specific binding to negative-charged cell membranes.

Reply: Thank you for the suggestions. Definitely, the zeta potential is an essential characterization of the nanoparticles. We have conducted the test and added the data to the manuscript. Specifically, the zeta potential is consistent on Day 0, Day 1 and Day 5 for the SnSNPs. We have added the data to the Figure S1a in the supplementary information file as below.

2. Please verify the following inconsistency. In Figure 2h, the y-axis is labeled as absorbance at 668 nm, but in the Method section, the wavelength is stated as 654 nm. While the absorbance peak of methylene blue may vary based on the experimental conditions, the inconsistency between the two values needs to be corrected.

Reply: Thank you for pointing this out. Indeed, the absorbance used for the MB assay is 668 nm. We have corrected our typo in the manuscript in line 504.

3. Line 132-133, Figure 2b is solely for the DPBF assay, while Figure 2e is for the MB assay, they should both be cited in line 133.

Reply: Thank you for the comment. We have revised the corresponding part and have cited both figures to the descriptions in line 133.

4. In Figure 3d, I would recommend the authors include more description regarding the parameters in the study design, such as wavelength of NIR, cell line information and US parameters. This additional information would provide readers with a better understanding of the experimental design.

Reply: Thank you for the suggestion. We have added detailed information regarding the experimental parameter of NIR (1.0 W cm^{-2}) and US (0.3 W cm^{-2}) in the illustration (Figure 3d).

Figure 3d. (d) Illustration of the in vitro SnSNPs@PEG-mediated SDT and groups of the assay.

5. The authors used low-intensity US in their in vitro assays, and although they have discussed the intensity concern in the Discussion section, it would be beneficial to mention it at the beginning of the corresponding description in the Results section to provide a clearer understanding for readers.

Reply: Thank you for your comment. We have added the following on this matter in the corresponding Result section: "It should be noted that for in vitro SDT, we used a low-intensity US to avoid US-induced cell detachment from tissue culture plate as this effect can significantly influence the results of cell viability assay."

6. In the biodistribution study, the authors noted that the most prominent accumulation of SnSNDs at the tumor site occurred at 12 h after injection. It would be helpful if the authors could provide an explanation for why they chose to wait 24 hours after i.v. injection before irradiation and treatment in their experiments?

Reply: We appreciate the reviewer's comment. In the biodistribution study, indeed, we observed the most prominent fluorescence signals of SnSNDs@PEG-Cy5 at the tumor site 12 hours post-injection. While this fluorescence may indicate the presence of SnSNPs at tumor region, it is important to note that it still needs time for their diffusion in the tumor tissue and uptake by tumor cells. Moreover, the SnSNPs tend to retain their accumulation at the tumor site between 12 and 24 hours. Considering these factors, we conducted the therapeutic assay based on SnSNPs with NIR irradiation at 24 hours after intravenous injection, which can ensure a sufficient and consistent presence of SnSNPs within tumor cells, allowing for the most effective and reliable therapeutic outcome.

7. The Discussion section is comprehensive and provides sufficient information in addition to the results. In the Discussion section, Line 400, it should not be low-intensity US as the authors used the same power of in-tube tests for in vivo studies.

Reply: We appreciate the reviewer's suggestion. We have revised our description as below: "Additionally, our in vivo assessment demonstrated the high biocompatibility of both NIR₈₀₈ and US (Figure S10, S14). It is worth mentioning that, despite the high biocompatibility of US in our tests and other reports⁵⁴, we used a relatively low-power-density US (0.3 W cm^{-2})⁵⁵ in the *in vitro* study to avoid interference of cell attachment to the plate."

8. There are a few minor points that could be improved in the manuscript. Firstly, the labeling color used for discriminating different groups could be made more consistent in Figure 3, Figure 6, and Figure 7. Additionally, the wavelength of NIR should be added to several descriptions where it is currently missing. Finally, there is a typo in Line 285, where "collagen" is misspelled.

Reply: Thank you for your suggestions. We have made the necessary changes, including relabeling the color for consistency and correcting the typo. We believe that these revisions will improve the quality of the manuscript.

REVIEWERS' COMMENTS

Reviewer #1 (Remarks to the Author):

I have no further technical comments. I still feel the novelty of this work is limited. However, I respect any decision made by the editor.

Reviewer #2 (Remarks to the Author):

The author has adequately addressed all the reviewer's concerns and therefore should be considered to be published in the current form.

RESPONSE TO REVIEWERS' COMMENTS

Reviewer #1 (Remarks to the Author):

I have no further technical comments. I still feel the novelty of this work is limited. However, I respect any decision made by the editor.

Response: We appreciate the reviewer's comments and feedback.

Reviewer #2 (Remarks to the Author):

The author has adequately addressed all the reviewer's concerns and therefore should be considered to be published in the current form.

Response: Thank you very much for your positive comments. We appreciate your valuable suggestions.